# Restless-UCB, an Efficient and Low-complexity Algorithm for Online Restless Bandits

**Siwei Wang[1], Longbo Huang[2], John C.S. Lui[3]**
[1]Department of Computer Science and Technology, Tsinghua University
wangsw2020@mail.tsinghua.edu.cn
[2]Institute for Interdisciplinary Information Sciences, Tsinghua University
longbohuang@mail.tsinghua.edu.cn
[3]Department of Computer Science and Engineering, The Chinese University of Hong Kong
cslui@cse.cuhk.edu.hk

## Abstract

We study the online restless bandit problem, where the state of each arm evolves according to a Markov chain, and the reward of pulling an arm depends on both the pulled arm and the current state of the corresponding Markov chain. In this paper, we propose Restless-UCB, a learning policy that follows the explore-then-commit framework. In Restless-UCB, we present a novel method to construct offline instances, which only requires $O(N)$ time-complexity ($N$ is the number of arms) and is exponentially better than the complexity of existing learning policy. We also prove that Restless-UCB achieves a regret upper bound of $\tilde{O}((N + M^3)T^{\frac{2}{3}})$, where $M$ is the Markov chain state space size and $T$ is the time horizon. Compared to existing algorithms, our result eliminates the exponential factor (in $M, N$) in the regret upper bound, due to a novel exploitation of the sparsity in transitions in general restless bandit problems. As a result, our analysis technique can also be adopted to tighten the regret bounds of existing algorithms. Finally, we conduct experiments based on real-world dataset, to compare the Restless-UCB policy with state-of-the-art benchmarks. Our results show that Restless-UCB outperforms existing algorithms in regret, and significantly reduces the running time.

## 1 Introduction

The restless bandit problem is a time slotted game between a player and the environment [50]. In this problem, there are $N$ arms (or actions), and the state of each arm $i$ evolves according to a Markov chain $M_i$, which makes one transition per time slot during the game (regardless of being pulled or not). At each time slot $t$, the player chooses one arm to pull. If he pulls arm $i$, he observes the current state $s_i(t)$ of $M_i$, and receives a random reward $x_i(t)$ that depends on $i$ and $s_i(t)$, i.e., $\mathbb{E}[x_i(t)] = r(i, s_i(t))$ for some function $r$. The goal of the player is to maximize his expected cumulative reward during the time horizon $T$, i.e., $\mathbb{E}[\sum_{t=1}^{T} x_{a(t)}(t)]$, where $a(t) \in [N]$ denotes the pulled arm at time step $t$.

Restless bandit can model many important applications. For instance, in a job allocation problem, an operator allocates jobs to $N$ different servers. The state of each server, i.e., the number of background jobs currently running at the server, can be modeled by a Markov chain, and it changes every time slot according to an underlying transition matrix [32, 20]. At each time slot, the operator allocates a job to one server, and receives the reward from that server, i.e., whether the job is completed, which depends on the current state of the server. At the same time, the operator can determine the current state of the chosen server based on its feedback. For servers that are not assigned jobs at the current

time slot, however, the operator does not observe their current state or transitions. The operator's objective is to maximize his cumulative reward in the $T$ time slots.

Another application of the restless bandit model is in wireless communication. In this scenario, a base-station (player) transmits packets over $N$ distinct channels to users. Each channel may be in "good" or "bad" state due to the channel fading condition, which evolves according to a two-state Markov chain [37, 40]. Every time, the player chooses one channel for packet transmission. If the transmission is successful, the player gets a reward of 1. The player also learns about the state of the channel based on receiver feedback. The goal of the player is to maximize his cumulative reward, i.e., deliver as many packets as possible, within the given period of time.

Most existing works on restless bandit focus on the offline setting, i.e., all parameters of the game are known to the player, e.g., [50, 48, 31, 8, 44]. In this setting, the objective is to search for the best policy of the player. In practice, one often cannot have the full system information beforehand. Thus, traditional solutions instead choose to solve the offline problem using empirical parameter values. However, due to the increasing sizes of the problem instances in practice, a small error on parameter estimation can lead to a large error, i.e., regret.

The online restless bandit setting, where parameters have to be learned online, has been gaining attention, e.g., [42, 13, 26, 23, 22, 33]. However, many challenges remain unsolved. First of all, existing policies may not perform close to the optimal offline one, e.g., [26] only considers the best policy that constantly pulls one arm. Second, for the class of Thompson Sampling based algorithms, e.g., [23, 22], theoretical guarantees are often established in the Bayesian setting, where the update methods can be computationally expensive when the likelihood functions are complex, especially for prior distributions with continuous support. Third, the existing policy with theoretical guarantee of a sublinear regret upper bound, i.e., colored-UCRL2 [33], suffers from an *exponential computation complexity* and a regret bound that is *exponential in the numbers of arms and states*, as it requires solving a set Bellman equations with an exponentially large space set.

In this paper, we aim to tackle the high computation complexity and exponential factor in the regret bounds for online restless bandit. Specifically, we consider a class of restless bandit problems with birth-death state Markov chains, and develop online algorithms to achieve a regret bound that is only polynomial in the numbers of arms and states. We emphasize that, birth-death Markov chains have been widely used to model real-world applications, e.g., queueing systems [24] and wireless communication [46], and are generalization of the two-state Markov chain assumption often made in prior works on restless bandits, e.g., [17, 28]. Our model can also be applied to many important applications, e.g., communications [3, 2], recommendation systems [30] and queueing systems [4].

The main contributions of this paper are summarized as follows:

- We consider a general class of online restless bandit problems with birth-death Markov structures and propose the Restless-UCB policy. Restless-UCB contains a novel method for constructing offline instances in guiding action selection, and only has an $O(N)$ complexity ($N$ is the number of arms), which is exponentially better than that of colored-UCRL2, the state-of-the-art policy with theoretical guarantee [33] for online restless bandits.

- We devise a novel analysis and prove that Restless-UCB achieves an $\tilde{O}((N + M^3)T^{\frac{2}{3}})$ regret, where $M$ is the Markov chain state space size and $T$ is the time horizon. Our bound improves upon existing regret bounds in [33, 22], which are exponential in $N, M$. The novelty of our analysis lies in the exploitation of the sparsity in general restless bandit problems, i.e., each belief state can only transit to $M$ other ones. This approach can also be combined with the analysis in [33, 22] to reduce the exponential factors in the regret bound to polynomial values (complexity remains exponential) in online restless bandit problems. Thus, our analysis can be of independent interest in online restless bandit analysis.

- We show that Restless-UCB can be combined with an efficient offline approximation oracle to guarantee $O(N)$ time-complexity and an $\tilde{O}(T^{\frac{2}{3}})$ approximation regret upper bound. Note that existing algorithms suffer from either an exponential complexity or no theoretical guarantee even with an efficient approximation oracle.

- We conduct experiments based on real-world datasets, and compare our policy with existing benchmarks. Our results show that Restless-UCB outperforms existing algorithms in both regret and running time.

## 1.1 Related Works

The offline restless bandit problem was first proposed in [50]. Since then, researchers concentrated on finding the exact best policy via index methods [50, 48, 28], i.e., first giving each arm an index, then choosing actions with the largest index and update their indices. However, index policies may not always be optimal. In fact, it has been shown that there exist examples of restless bandit problems where no index policy achieves the best cumulative reward [50]. [35] further shows that finding the best policy of any offline restless bandit model is a PSPACE-hard problem. As a result, researchers also worked on finding an approximate policy of restless bandit [27, 17].

There have also been works on online restless bandit. One special case is the stochastic multi-armed bandit [7, 25] in which the arms all have singe-state Markov chains. Under this setting, the best offline policy is to choose the action with the largest expected reward forever. Researchers also propose numerous policies to solve the online problem, classical algorithms include UCB policy [5] and the Thompson Sampling policy [43].

[42, 13, 26] considered the online restless bandit model with weak regret, i.e., comparing with single action policies (which is similar as the best policy in stochastic multi-armed bandit model), and they proposed UCB-based policies. Specifically, the algorithms choose to pull a single arm for a long period, so that the average reward during this period is close to the actual average reward of always pulling this single arm. Based on this fact, they established the upper confidence bounds for every arm, and showed that always choosing the action with the largest upper confidence bound achieves an $O(\log T)$ weak regret.

To solve the general online restless bandit problem without policy constraints, [33] showed that the problem can be regarded as a special online reinforcement learning problem [41]. In this setting, prior works adapted the idea of UCB and Thompson Sampling, and proposed policies with regret upper bound $O(D\sqrt{T})$, where $D$ is the diameter of the game [21, 34, 1]. Based on these approaches, people proposed UCB-based policies, e.g., colored-UCRL2 [33], and Thompson Sampling policies, e.g., [23, 22] for the online restless bandit problem. Colored-UCRL2 directly applies the UCRL policy in [21], and leads to an $O(D\sqrt{T})$ regret upper bound. Also, to search for a best problem instance within a confidence set, it needs to solve a set of Bellman equations with an exponentially large space set, resulting in an exponential time complexity. To avoid this exponential time cost, [22] adapted the idea of Thompson Sampling in reinforcement learning [1], and achieves a Bayesian regret upper bound $O(D\sqrt{T})$. However, it requires a complicated method for updating the prior distributions to posterior ones when the likelihood functions are complex, especially for prior distributions with continuous support. In addition to the large time complexity, another challenge is that, the diameter $D$ of the game is usually exponential with the size of the game, leading to a large regret bound. [23] considered a different setting, i.e., the episodic one, in which the game always restarts after $L$ time steps. This way, it avoided the $D$ factor in the regret bound and achieved an $O(L\sqrt{T})$ regret.

A related topic of restless bandit is non-stationary bandit problem [15, 9], in which the expected reward of pulling each arm may vary across time. In non-stationary bandit, people work on the settings with limited number of breakpoints (the time steps that the expected rewards change) [15] or limited variation on the expected rewards [9]. The main difference is that non-stationary bandit problem does not assume any inner structure about how the expected rewards vary. In restless bandit, we assume that they follow a Markov chain structure, and focus on learning this structure out by observing more information about it (i.e., except for the received reward, we also observe the current state of the chosen action). Therefore, the algorithm and analysis for restless bandit can be very different with those for non-stationary bandit.

## 2   Model Setting

Consider an online restless bandit problem $\mathcal{R}$ which has one player (decision maker) and $N$ arms (actions) $\{1, \cdots, N\}$. Each arm $i \in \{1, \cdots, N\}$ is associated with a Markov chain $M_i$. All the Markov chains $\{M_i, i = 1, 2, ..., N\}$ have the same state space $S = \{1, 2, \cdots, M\}$,[1] but may have different transition matrices $\{\boldsymbol{P}_i, i = 1, 2, ..., N\}$ and state-dependent rewards $\{r(i, s), \forall i, s\}$ that

are unknown to the player. The initial states of the arms are denoted by $\boldsymbol{s}(0) = [s_1(0), \cdots, s_N(0)]$. The game duration is divided into $T$ time steps. In each time $t$, the player chooses an arm $a(t) \in [N]$ to play. We assume without loss of generality that there is a default arm 0, whose existence does not influence the theoretical results in this paper, and it is only introduced to simplify the proofs.

If the chosen arm $a(t)$ is not the default one, i.e., $a(t) > 0$, this action gives a reward $x(t) \in [0, 1]$, which is an independent random variable with expectation $r(a(t), s_{a(t)}(t))$, where $s_{a(t)}(t)$ is the state of $M_{a(t)}$ at time $t$. On the other hand, pulling arm 0 always results in a reward of 0. In every time step, the Markov chain of each arm makes a transition according to its transition matrix, regardless of whether it is pulled or not. However, the player only observes the current state and reward of the chosen arm. The current states of the rest of the arms are unknown. The goal of the player is to design an online learning policy to maximize the total expected reward during the game.

We use *regret* to evaluate the efficiency of the learning policy, which is defined as the expected gap between the offline optimal, i.e., the best policy under the full knowledge of all transition matrices and reward information, and the cumulative reward of the arm selecting algorithm. Specifically, let $\mu(\pi, \mathcal{R})$ denote the expected average reward under policy $\pi$ for problem $\mathcal{R}$, i.e., $\mu(\pi, \mathcal{R}) = \lim_{T \to \infty} \frac{1}{T} \sum_{t=1}^{T} \mathbb{E}[x^\pi(t)]$, where $x^\pi(t)$ is the random reward at time $t$ when applying policy $\pi$ during the game. Then, we define the optimal average reward $\mu^*(\mathcal{R})$ as $\mu^*(\mathcal{R}) = \sup_\pi \mu(\pi, \mathcal{R})$. The regret of policy $\pi$ is then defined as $Reg(T) = T\mu^*(\mathcal{R}) - \sum_{t=1}^{T} \mathbb{E}[x^\pi(t)]$.

Next, we state the assumptions made in the paper.

**Assumption 1.** *For any action $i$, we have $r(i, j) \geq r(i, k)$ for states $j < k$.*

This assumption is common in real-world applications, and is widely adopted in the restless bandit literature, e.g., [28, 2]. For instance, in job allocation, a busy server has a larger probability of dropping an incoming job than an idle server. Another example is in wireless communication, where transmitting in a good channel has a higher success probability than in a bad channel.

The next assumption is that all Markov chains have a birth-death structure, which are common in a wide range of problems including queueing systems [24] and wireless communications [46]. We also note that the birth-death Markov chain generalizes the two-state Markov chain assumption which was used in prior works of restless bandit [17, 28].

**Assumption 2.** $P_i(j, k) = 0$ *for any action $i$ and state $|j - k| > 1$, where $P_i(j, k)$ is the probability that $M_i$ transits from state $j$ to $k$ in one time step.*

The following assumption is a generalization of the positive-correlated assumption often made in the restless bandit literature, e.g., [2, 45].

**Assumption 3.** *For any action $i$, state $1 \leq k \leq M - 1$, we have that $P_i(k, k+1) + P_i(k+1, k) \leq 1$.*

Finally, we assume that for any state $j$, the probability of going to any neighbor state is lower bounded by a constant. This assumption is rather mild, especially when we only have a finite number of states.

**Assumption 4.** *For any action $i$, state $|j - k| \leq 1$, we have that $P_i(j, k) \geq c_1$ for some constant $c_1 \in (0, 1)$.*

Under these assumptions, it is easy to verify that the Markov chains are ergodic. We thus denote the unique stationary distribution of $M_i$ by $\boldsymbol{d}^{(i)} = [d_1^{(i)}, \cdots, d_M^{(i)}]$, and denote $d_{\min} \triangleq \min_{i,k} d_k^{(i)}$. For each transition matrix $\boldsymbol{P}_i$, we also define a neighbor space $\mathcal{P}_i$ of transition matrices as:

$$\mathcal{P}_i = \left\{ \tilde{\boldsymbol{P}}_i : \forall |j - k| \leq 1, |\tilde{P}_i(j, k) - P_i(j, k)| \leq \frac{2c_1}{3} \right\}.$$

Notice that any Markov chain $\tilde{M}_i$ with transition matrices $\tilde{\boldsymbol{P}}_i \in \mathcal{P}_i$ must be ergodic. Thus, the absolute value of the second largest eigenvalues of $\tilde{\boldsymbol{P}}_i$, denoted by $\lambda_{\tilde{\boldsymbol{P}}_i}$, is smaller than 1, which means $\lambda^i \triangleq \sup_{\tilde{\boldsymbol{P}}_i \in \mathcal{P}_i} \lambda_{\tilde{\boldsymbol{P}}_i} < 1$ and $\lambda_{\max} \triangleq \max_i \lambda^i < 1$.

## 3 Restless-UCB Policy

In this section, we present our Restless-UCB policy, whose pseudo-code is presented in Algorithm 1.

---

**Algorithm 1** Restless-UCB Policy

---

1: **Input:** Time horizon $T$, learning function $m(T)$.
2: **for** $i = 1, 2, \cdots, N$ **do**
3:    Choose arm $i$ until there are $m(T)$ times we observe $s_i(t) = k$ for all states $k$.
4: **end for**
5: Let $\hat{P}_i(j, k)$'s and $\hat{r}(i, k)$'s be the empirical values of $P_i(j, k)$'s and $r(i, k)$'s.
6: Construct instance $\mathcal{R}'$ with $r'(i, k) = \hat{r}(i, k) + rad(T)$, $P_i'(k, k+1) = \hat{P}_i(k, k+1) - rad(T)$, $P_i'(k, k) = \hat{P}_i(k, k)$, $P_i'(k, k-1) = \hat{P}_i(k, k-1) + rad(T)$. Specifically, $P_i'(1, 1) = \hat{P}_i(1, 1) + rad(T)$ and $P_i'(M, M) = \hat{P}_i(M, M) - rad(T)$.
7: Find the optimal policy $\pi^{*\prime}$ for problem $\mathcal{R}'$, i.e., $\pi^{*\prime} = \texttt{Oracle}(\mathcal{R}')$.
8: **while true do**
9:    Follow $\pi^{*\prime}$ for the rest of the game.
10: **end while**

---

Restless-UCB contains two phases: (i) the exploration phase (lines 2-4) and (ii) the exploitation phase (lines 5-10). The goal of the exploration phase is to learn the parameters $\{\boldsymbol{P}_i, i = 1, 2, ..., N\}$ and $\{r(i, s), \forall i, s\}$ as accurate as possible. To do so, Restless-UCB pulls each arm until there are sufficient observations, i.e., for any action $i$ and state $k$, we observe the next transition and the given reward for at least $m(T)$ number of times ($m(T)$ to be specified later). Once there are $m(T)$ observations, the empirical values of $\{\boldsymbol{P}_i, i = 1, 2, ..., N\}$ and $\{r(i, s), \forall i, s\}$ (represented by $\hat{P}_i(j, k)$ and $\hat{r}(i, k)$) have a bias within $rad(T) \triangleq \sqrt{\frac{\log T}{2m(T)}}$ with high probability [19]. The key here is to choose the right $m(T)$ to balance accuracy and complexity.

In the exploitation phase, we first use an offline oracle $\texttt{Oracle}$ (using oracle is a common approach in bandit problems [10, 11, 49]) to construct the optimal policy for our estimated model instance based on empirical data, and then apply this policy for the rest of the game. The key to guarantee good performance of our algorithm is that, instead of using the empirical data directly, in Restless-UCB, *we carefully construct an offline problem instance to guide our policy search*. Specifically, we use the *upper* confidence bound values for $P_i(k, k-1)$'s and $r(i, k)$'s, the *lower* confidence bounds for $P_i(k, k+1)$'s, and the empirical values for $P_i(k, k)$'s. As shown in line 6 of Algorithm 1, we set $r'(i, k) = \hat{r}(i, k) + rad(T)$, $P_i'(k, k+1) = \hat{P}_i(k, k+1) - rad(T)$, $P_i'(k, k) = \hat{P}_i(k, k)$, $P_i'(k, k-1) = \hat{P}_i(k, k-1) + rad(T)$ in the estimated offline instance $\mathcal{R}'$. This method allows us to use $O(N)$ complexity to construct a good offline instance, which is greatly better than the exponential cost in [33].

Next, we view the offline restless bandit instance as a MDP. The state of the MDP (referred to as belief state in POMDP [16]), is defined as to be $z = \{(s_i, \tau_i)\}_{i=1}^N$, where $s_i$ is the last observed state of $M_i$, $\tau_i$ is the number of time steps elapsed since the last time we observe $M_i$, and the action set is $\{1, 2, \cdots, N\}$. Once we choose action $i$ under belief state $z$, the belief state will transit to $z_k^i$ with probability $p_k$, where $p_k$ equals to the $k$-th term in vector $\boldsymbol{e}_{s_i} \boldsymbol{P}_i^{\tau_i}$ ($\boldsymbol{e}_{s_i}$ represents the one hot vector with only the $s_i$-th term equals to 1 and the rest are 0), and $z_k^i = \{(s_j, \tau_j + 1)\}_{j \neq i} \cup \{k, 1\}$, i.e., $\{s_i, \tau_i\}$ is updated by $\{k, 1\}$ according to the observation, while other actions only have their $\tau_j$ values increase by one. We then use $\texttt{Oracle}$ to find out the optimal policy $\pi^{*\prime}$ for the empirical offline instance $\mathcal{R}'$. After that, Algorithm 1 uses policy $\pi^{*\prime}$ for the rest of the game, even though the actual model is $\mathcal{R}$ rather than $\mathcal{R}'$.

Note that by choosing $m(T) = o(T)$, the major part of the game will be in the exploitation phase, whose size is close to $T$. Thus, the regret in the exploitation phase is about $T(\mu(\pi^*, \mathcal{R}) - \mu(\pi^{*\prime}, \mathcal{R}))$, where $\pi^*$ is the best policy of the origin problem $\mathcal{R}$. To bound the regret, we divide the gap into two parts and analyze them separately, i.e., $T[\mu(\pi^*, \mathcal{R}) - \mu(\pi^{*\prime}, \mathcal{R})] = T[\mu(\pi^*, \mathcal{R}) - \mu(\pi^{*\prime}, \mathcal{R}')] + T[\mu(\pi^{*\prime}, \mathcal{R}') - \mu(\pi^{*\prime}, \mathcal{R})]$.

Roughly speaking, our estimation in the exploration phase makes the probability of transitioning to a lower index state (which has a higher expected reward) larger in any Markov chain $M_i$. Thus, the corresponding estimated reward is also larger. This way, one can guarantee that with high probability, the average reward of applying $\pi^{*\prime}$ in $\mathcal{R}'$ is larger than that of applying $\pi^*$ in the original model $\mathcal{R}$, i.e., the first term $T[\mu(\pi^*, \mathcal{R}) - \mu(\pi^{*\prime}, \mathcal{R}')]$ is less than or equal to zero. The second term is the average reward gap between applying the same policy in different problem instance $\mathcal{R}$ and $\mathcal{R}'$. Since

our estimation ensures that $\mathcal{R}'$ and $\mathcal{R}$ only has a bounded gap, this term can also be bounded. The regret bound and theoretical analysis are shown in details in Section 4.

We emphasize that although Restless-UCB has a similar form as an "explore-then-commit" policy, e.g., [29, 36, 14], the key novelty of our scheme lies in the method of constructing offline problem instances from empirical data. Our method also only requires $O(N)$ time to search for a better problem instance within the confidence set, which greatly reduces the running time of our algorithm. In contrast, existing algorithms, e.g., [33], take exponential time for this step and incur an exponential (in $N$) implementation cost.

As described before, our method chooses to use upper (lower) confidence bounds of the transition probabilities in $\boldsymbol{P}_i$. If observations for different arms are interleaved with each other, it will be very difficult to utilize them directly for updating the confidence interval of $\boldsymbol{P}_i$, since they are observations of transition matrix $\boldsymbol{P}_i^\tau$ with $\tau > 1$ but not $\boldsymbol{P}_i$. Indeed, according to [18], it is already difficult to calculate the $\tau$-th roots of stochastic matrices, let alone finding the confidence intervals. Therefore, we construct Markov chain $M_i'$ in the offline instance $\mathcal{R}'$ by continuously pulling a single arm $i$ for long time. This is why we choose to use an "explore-then-commit" framework instead of a UCRL framework.

## 4 Theoretical Analysis

In this section, we present our main theoretical results. The complete proofs are referred to the Appendix A in supplementary file.

### 4.1 Restless-UCB with Efficient Offline Oracles

We first consider the case when there exists an efficient `Oracle` for the offline restless bandit problem.

**Theorem 1.** *If `Oracle` returns the optimal policy, then under Assumptions 1, 2, 3 and 4, the regret of Restless-UCB with $m(T) = T^{\frac{2}{3}}$ in an online restless bandit problem $\mathcal{R}$ satisfies:*

$$Reg(T) = \tilde{O}\left(\left(\frac{N}{d_{\min}} + \frac{M^3}{(1 - \lambda_{\max})^2}\right) T^{\frac{2}{3}}\right).$$

Although our setting focuses on birth-death Markov state transitions, we note that it is still general and applies to a wide range of important applications, e.g., communications [3, 2], recommendation systems [30] and queueing systems [4]. Focusing on this setting allows us to design algorithms with much lower complexity, which can be of great interest in practice. Existing low-complexity policies, though applicable to general MDP problems, do not have theoretical guarantees. For example, the UCB-based policies in [42, 13, 26] suffer from a $\Theta(T)$ regret (although their weak regret upper bound is $O(\log T)$), while the Thompson Sampling policy only has a sub-linear regret upper bound in Bayesian setting. The colored-UCRL2 policy [33], which possesses a sub-linear regret bound of $O(D\sqrt{T})$ with respect to $T$, suffers from an exponential implementation complexity (in $N$), even with an efficient oracle. Our Restless-UCB policy only requires a polynomial complexity (refer to Line 6 in Algorithm 1) with an efficient offline oracle, and achieves a rigorous sub-linear regret upper bound.

The reason why the regret bound of Restless-UCB is slightly worse than colored-UCRL2 is because the observations in the exploitation phase are not used for updating the parameters of the game (the observations for different arms in the exploitation phase are interleaved with each other and are hard to be used as we discussed before). This means that only $\tilde{O}(T^{\frac{2}{3}})$ observations are used in estimating the offline problem instance, resulting in a bias of $\tilde{O}(T^{-\frac{1}{3}})$ according to [19]. Colored-UCRL2 tackles this problem (i.e., utilize all the observations) by updating transition vectors for all the possible $(s_i, \tau_i)$, and finding a policy based on all these transition vectors. As a result, it needs to work with an exponential space set and results in an exponential time complexity. We instead sacrifice a little on the regret to obtain a significant reduction in the implementation complexity. We also emphasize that the colored-UCRL2 policy cannot simplify its implementation even under our assumptions, due to its need to compute the best policy based on all transition vectors for all (exponentially many) $(s_i, \tau_i)$ pairs. Moreover, the factor in the Restless-UCB's regret bound is $\frac{N}{d_{\min}} + \frac{M^3}{(1-\lambda_{\max})^2}$,

which is polynomial with $N, M$, and is *exponentially* better than the factor $D$ in regret bounds of colored-UCRL2 [33] or Thompson Sampling [22], which is the diameter of applying a learning policy to the problem and is exponential in $N, M$. In Appendix B of the supplementary file, we also prove that our analysis can be adopted to reduce the $D$ factor in their regret bounds to polynomial values. This shows that our analysis approach can be of independent interest for analyzing online restless bandit problems.

Now we highlight two key ideas of our theoretical analysis, each summarized in one lemma. They enable us to achieve polynomial time complexity and reduce the exponential factor in regret bounds.

**Lemma 1.** *Conditioning on event $\mathcal{E}$, we have $\mu(\pi^*, \mathcal{R}) \leq \mu(\pi^{*\prime}, \mathcal{R}')$. Here*

$$\mathcal{E} = \{\forall i, |j - k| \leq 1, |P_i(j, k) - \hat{P}_i(j, k)| \leq rad(T), |r(i, k) - \hat{r}(i, k)| \leq rad(T)\}.$$

**Remark 1.** *Conditioning on event $\mathcal{E}$, which is guaranteed to happen with high probability, the probability of transitioning to a lower index state (which has a higher expected reward) in $\mathcal{R}'$ is always larger than that one in $\mathcal{R}$. Lemma 1 shows that we can obtain a higher average reward in $\mathcal{R}'$ than in $\mathcal{R}$. This implies that we can efficiently (with $O(N)$ complexity) construct a better instance within the confidence set, which is an important step in analyzing restless MDPs, e.g., [33], whereas the prior work takes an exponential time for the construction.*

**Lemma 2.** *Conditioning on event $\mathcal{E}$, $\mu(\pi^{*\prime}, \mathcal{R}') - \mu(\pi^{*\prime}, \mathcal{R}) = \tilde{O}\left(\frac{M^3}{(1-\lambda_{\max})^2} T^{-\frac{1}{3}}\right)$.*

**Remark 2.** *Existing results in [33, 22] treat the restless bandit game as a general MDP. Doing so results in the factor $D$ in their regret upper bounds ($D$ is the diameter of the game and is exponential in the game size). In our case, the factor $\frac{M^3}{(1-\lambda_{\max})^2}$ in Lemma 2 is polynomial with the the game size. This improvement comes from a novel exploitation in the sparsity in general restless bandit problems, i.e., each belief state can only transit to $M$ other ones. This novel result can also help to reduce the exponential factors in previous regret bounds, e.g., [33, 22], to polynomial ones in general restless bandit problems, though their complexities remain exponential (see details in Appendix B of the supplementary file), and can be of independent interest in analyzing online restless bandit problems.*

### 4.2 Restless-UCB with Efficient Offline Approximate Oracles

In this section, we show how Restless-UCB can be combined with approximate oracles to achieve good regret performance. In practical applications, finding the optimal policy of the offline problem is in general NP-hard [35] and one can only obtain efficient approximate solutions [27, 17]. As a result, how to perform learning efficiently and achieve low regret when `Oracle` can only return an approximation policy, e.g., [10, 11, 49], is of great interest and importance in designing low-complexity and efficient learning policies.

The definitions of approximate policies and approximation regret is given below.

**Definition 1.** *For an offline restless bandit instance $\mathcal{R}$, an approximate policy $\tilde{\pi}$ with approximate ratio $\lambda > 0$ satisfies that $\mu(\tilde{\pi}, \mathcal{R}) \geq \lambda \mu^*(\mathcal{R})$.*

**Definition 2.** *For an online restless bandit instance $\mathcal{R}$, the approximation regret with approximate ratio $\lambda > 0$ for learning policy $\pi$ is defined as $Reg(T, \lambda) = \lambda \mu^*(\mathcal{R}) - \sum_{t=1}^{T} \mathbb{E}[x^\pi(t)]$.*

**Theorem 2.** *If `Oracle` returns an approximate policy with ratio $\lambda$, then under Assumptions 1, 2, 3 and 4, the approximation regret (with approximate ratio $\lambda$) of Restless-UCB with $m(T) = T^{\frac{2}{3}}$ in an online restless bandit problem $\mathcal{R}$ is upper bounded by $\tilde{O}(T^{\frac{2}{3}})$.*

Theorem 2 shows that Restless-UCB can be combined with approximate policies for the offline problem to achieve good performance, i.e., it can reduce the time complexity by applying an approximate oracle. This feature is not possessed by other policies such as colored-UCRL2 [23] or Thompson Sampling [22]. Specifically, colored-UCRL2 needs to solve a set of Bellman equations with exponential size to find out the better instance within the confidence set, thus using an approximate policy leads to a similar approximation regret but cannot reduce the time complexity. Thompson Sampling, on the other hand, can only apply the approximate approach in the Bayesian setting [47].

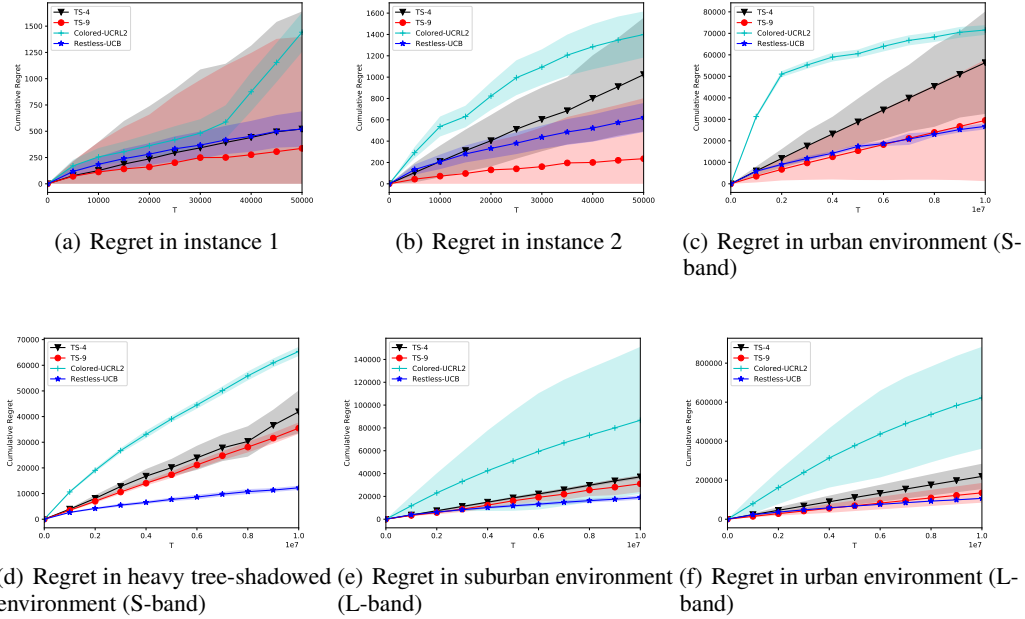

(a) Regret in instance 1

(b) Regret in instance 2

(c) Regret in urban environment (S-band)

(d) Regret in heavy tree-shadowed environment (S-band)

(e) Regret in suburban environment (L-band)

(f) Regret in urban environment (L-band)

Figure 1: Experiments: Comparison of regrets of different algorithms

## 5 Experiments

In this section, we present some of our experimental results. In all these experiments, we use the offline policy proposed by [28] as the offline oracle of restless bandit problems.

### 5.1 Experiments on Constructed Instance

We consider two problem instances, each instance contains two arms and each arm evolves according to a two-state Markov chain. In both instances, $r(i, 2) = 0$ for any arm $i$, and all Markov chains start at state 2. In problem instance one, $r(1, 1) = 1$, $r(2, 1) = 0.8$, $P_1(1, 1) = 0.7$, $P_1(2, 2) = 0.8$, $P_2(1, 1) = 0.5$ and $P_2(2, 2) = 0.6$. In problem instance two, $r(1, 1) = 0.8$, $r(2, 1) = 0.4$, $P_1(1, 1) = 0.7$, $P_1(2, 2) = 0.9$, $P_2(1, 1) = 0.7$ and $P_2(2, 2) = 0.5$.

We compare three different algorithms, including Restless-UCB, state-of-the-art colored-UCRL2 [33] and Thompson Sampling policies with different priors, TS-9 and TS-4 [22]. In TS-9, the prior distributions of transition probability of Markov chain $M_i$ are the uniform one on $[0.1, 0.2, 0.3, 0.4, 0.5, 0.6, 0.7, 0.8, 0.9]^2$ (for the values $P_i(1, 1)$ and $P_i(2, 2)$), and the prior distributions of different Markov chains are independent. In TS-4, the prior support is $[0.2, 0.4, 0.6, 0.8]^2$ instead.

The regrets of these algorithms are shown in Figures 1(a) and 1(b), which take average over 1000 independent runs. The expected regrets of Restless-UCB are smaller than TS-4 and colored-UCRL2, but larger than TS-9. This is because the support of prior distribution in TS-9 contains the real problem instance. As a result, its samples equal to the real problem instance with high probability. However, from the expected regrets of TS-4, one can see that when the support of its prior distribution is not close to the real instance, its expected regret grows linearly as $T$ increases. Besides, compare with Restless-UCB, TS policy has a much larger variance on the regret, due to the high degree of randomness on the samples. Therefore, Restless-UCB is more robust against inaccurate estimation in the prior distributions, and has a more reliable theoretical guarantee due to its low variance on regret.

We also compare the average running times of the different algorithms. In this experiment, $T = 500000$. The four problem instances contain $N = 2, 3, 4, 5$ arms and each arm has a two-state Markov chain. The results in Table 1 take average over 50 runs of a single-threaded program (running on an Intel E5-2660 v3 workstation with 256GB RAM). They show that Restless-UCB policy is

much more efficient when there are more arms (particularly compared to colored-UCRL2). The time complexity of colored-UCRL2 grows exponentially as $N$ grows up, while the time complexity of Restless-UCB only has a minimal increase.

Table 1: Average running times of different algorithms

| Algorithm | 2 arms | 3 arms | 4 arms | 5 arms |
|---|---|---|---|---|
| Restless-UCB | 4s | 5s | 5s | 5s |
| TS-9 | 4s | 6s | 8s | 10s |
| TS-4 | 3s | 4s | 6s | 9s |
| Colored-UCRL2 | 5s | 326s | 8892s | 129387s |

## 5.2 Experiments with Real Data Set

We also use real datasets to compare the behavior of different algorithms.

Here we use the wireless communication dataset in [37]. It is a setting on digital video broadcasting satellite services to handheld devices via land mobile satellite. [37] provided the parameters of two-state Markov chain representations on the channel model in three different environments, including urban, suburban and heavy tree-shadowed environments. In this experiment, different elevation angles of antenna are represented as arms, and different elevation angles correspond to different channel parameters, including the transition matrices and propagation impairments. Our goal is to correctly transmit as many data packets as possible within a time horizon $T$. We use the transition probability matrices in Tables IV and VI in [37]. We also use the average direct signal mean given in Tables III and V in [37] as the expected reward. In Figure 1(c), we consider communicating via S-band under the urban environment, and one can choose the elevation angle to be either $40°$ or $80°$. In Figure 1(d), we consider communicating via S-band under the heavy tree-shadowed environment, and one can choose the elevation angle to be either $40°$ or $80°$. In Figure 1(e), we consider communicating via L-band under the suburban environment, and one can choose the elevation angle to be $50°$, $60°$ or $70°$. In Figure 1(f), one consider communicating via L-band under the urban environment, and we can choose the elevation angle to be either $10°$, $20°$, $30°$ or $40°$. All of these results take average over 200 independent runs.

One can see that Restless-UCB performs the best in all these experiments, it achieves the smallest expected regret and the smallest variance on regret. As mentioned before, the TS policy suffers from a linear expected regret since its support does not contain the real transition matrices, and it has a large variance on regret at the same time. Although colored-UCRL2 achieves a sub-linear regret, it suffers from a large constant factor and performs worse than Restless-UCB. When there are more arms (see Figures 1(e) and 1(f)), colored-UCRL2 also suffers from a large variance on regret. These results demonstrate the effectiveness of Restless-UCB.

## 6 Conclusion

In this paper, we propose a low-complexity and efficient algorithm, called Restless-UCB, for online restless bandit. We show that Restless-UCB achieves a sublinear regret upper bound $\tilde{O}(T^{\frac{2}{3}})$, with a polynomial time implementation complexity. Our novel analysis technique also helps to reduce both the time complexity of the policy and the exponential factor in existing regret upper bounds. We conduct experiments based on real-world datasets to show that Restless-UCB outperforms existing benchmarks and has a much shorter running time.

## Broader Impact

Online restless bandit model has found applications in many important areas such as wireless communications [3, 2], recommendation systems [30] and queueing systems [4]. Existing results face challenges including exponential implementation-complexity and regret bounds that are exponential in the size of the game [33, 22, 23]. Our Restless-UCB algorithm offers a novel approach that enjoys $O(N)$ time-complexity to implement, which greatly reduces the running time in real applications.

Moreover, our analysis reduces the exponential factor in the regret upper bound to a polynomial one. Our work contributes to designing low-complexity and efficient learning policies for online restless bandit problem and can likely find applications in a wide range of areas.

## Acknowledgments and Disclosure of Funding

The work of Siwei Wang and Longbo Huang was supported in part by the National Natural Science Foundation of China Grant 61672316, the Zhongguancun Haihua Institute for Frontier Information Technology and the Turing AI Institute of Nanjing.

The work of John C.S. Lui is supported in part by the GRF 14201819.

## Footnotes

[1]This is not restrictive and is only used to simplify notations. Our analysis still works in the case where the state space $S_i$ of Markov chain $M_i$ satisfies that $|S_i| \leq M$.

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
