[Supplementary Material]

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

# Supplementary Material

## A    Proof of Theorem 1

In this section, we first propose the proofs of our two key lemmas, i.e., Lemma 1 and Lemma 2. Then we state other lemmas and facts that are helpful in the proof of Theorem 1. Finally, we show the complete proof of Theorem 1.

In the following, we concentrate on the case that $rad(T) \leq \frac{c_1}{3}$. If $rad(T) \geq \frac{c_1}{3}$, then $T = \tilde{O}(\frac{1}{c_1^3})$, which implies that the regret is at most $\tilde{O}(\frac{1}{c_1^3})$.

### A.1    Proof of Lemma 1

We first introduce a useful definition:

**Definition 3.** *For two vectors $\boldsymbol{v}$ and $\boldsymbol{v}'$ with $M$ dimensions, $\boldsymbol{v} \gtrsim \boldsymbol{v}'$ if for all $k \leq M$, $\sum_{j=1}^{k} v_j \geq \sum_{j=1}^{k} v_j'$.*

Based on this definition, we have the following lemmas.

**Lemma 3.** *Under Assumptions 2, 4 and conditioning on event $\mathcal{E}$, we have that $\boldsymbol{P}_i'(k) \gtrsim \boldsymbol{P}_i(k)$, where $\boldsymbol{P}_i'(k)$ and $\boldsymbol{P}_i(k)$ represent the transition vectors of arm $i$ under state $k$ in our estimated model $\mathcal{R}'$ and origin model $\mathcal{R}$, respectively.*

*Proof.* Note that there are only three non-zero values in $\boldsymbol{P}_i(k)$ (and $\boldsymbol{P}_i'(k)$, respectively), i.e., $P_i(k, k-1)$, $P_i(k, k)$ and $P_i(k, k+1)$ ($P_i'(k, k-1)$, $P_i'(k, k)$ and $P_i'(k, k+1)$, respectively). Then we only need to prove that conditioning on event $\mathcal{E}$, we have that $P_i'(k, k+1) \leq P_i(k, k+1)$ and $P_i'(k, k-1) \geq P_i(k, k-1)$. This is given by definition of $\mathcal{E}$ and $\boldsymbol{P}_i'(k)$'s directly.    □

**Lemma 4.** *Under Assumptions 2, 3, for any arm $i$ and state $k$, we have that $\boldsymbol{P}_i(k) \gtrsim \boldsymbol{P}_i(k+1)$.*

*Proof.* Note that $P_i(k, k-1) \geq 0 = P_i(k+1, k-1)$ and $P_i(k, k-1) + P_i(k, k) + P_i(k, k+1) = 1 \geq 1 - P_i(k+1, k+2) = P_i(k+1, k-1) + P_i(k+1, k) + P_i(k+1, k+1)$. Thus, the only thing we need to prove is that $P_i(k, k-1) + P_i(k, k) \geq P_i(k+1, k)$.

Since we have

$$P_i(k, k-1) + P_i(k, k) = 1 - P_i(k, k+1) \geq P_i(k+1, k),$$

where the inequality is because of Assumption 3, we finish the proof of this lemma.    □

**Lemma 5.** *Under Assumptions 2, 4 and conditioning on event $\mathcal{E}$, for any arm $i$ and probability vector $\boldsymbol{v}$, we have that $\boldsymbol{v} \boldsymbol{P}_i' \gtrsim \boldsymbol{v} \boldsymbol{P}_i$.*

*Proof.* Define $(\boldsymbol{v})_1^k = \sum_{j=1}^{k} v_j$, then we only need to prove that for any arm $i$ and state $k$, we have that $(\boldsymbol{v} \boldsymbol{P}_i')_1^k \geq (\boldsymbol{v} \boldsymbol{P}_i)_1^k$.

Note that

$$
\begin{aligned}
(\boldsymbol{v} \boldsymbol{P}_i')_1^k &= \sum_{j=1}^{M} (v_j \boldsymbol{P}_i'(j))_1^k \\
&= \sum_{j=1}^{M} v_j (\boldsymbol{P}_i'(j))_1^k \\
&\geq \sum_{j=1}^{M} v_j (\boldsymbol{P}_i(j))_1^k \qquad (1) \\
&= (\boldsymbol{v} \boldsymbol{P}_i)_1^k,
\end{aligned}
$$

where Eq. (1) is because that conditioning on event $\mathcal{E}$, we have that $\boldsymbol{P}_i'(k) \gtrsim \boldsymbol{P}_i(k)$ (Lemma 3).    □

---
**Algorithm 2** $\hat{\pi}^*$ based on $\pi^*$
---
1: **Init:** The real belief state is $z' = \{(s'_i, \tau'_i)\}_{i=1}^N$, and set the virtual belief state $z = z'$.
2: **while true do**
3:     Choose action $a(t)$ as $\pi^*$ choose under $z$, and observe state $s'_i(t)$, reward $r'(a(t), s'_i(t))$.
4:     Set $\boldsymbol{v}' = \boldsymbol{e}_{s'_{a(t)}}(\boldsymbol{P}'_{a(t)})^{\tau_{a(t)}}$ and $\boldsymbol{v} = \boldsymbol{e}_{s_{a(t)}}(\boldsymbol{P}_{a(t)})^{\tau_{a(t)}}$.
5:     Update the $a(t)$-th term in $z$ to be $(s_i(t), 1)$, where $s_i(t) = \mathsf{Correspond}(a(t), \boldsymbol{v}, \boldsymbol{v}', s'_i(t))$. For other terms $j \neq a(t)$, set $\tau_j = \tau_j + 1$. Observes virtual reward $r(a(t), s_i(t))$.
6:     Update the $a(t)$-th term in $z'$ to be $(s'_i(t), 1)$, for other terms $j \neq a(t)$, set $\tau'_j = \tau'_j + 1$.
7: **end while**
---

**Lemma 6.** *Under Assumptions 2, 3 and conditioning on event $\mathcal{E}$, for any arm $i$ and any probability vectors $\boldsymbol{v}$ and $\boldsymbol{v}'$ such that $\boldsymbol{v} \gtrsim \boldsymbol{v}'$, we have that $\boldsymbol{v}\boldsymbol{P}_i \gtrsim \boldsymbol{v}'\boldsymbol{P}_i$.*

*Proof.* We only need to prove that for any arm $i$ and state $k$, we have that $(\boldsymbol{v}\boldsymbol{P}_i)_1^k \geq (\boldsymbol{v}'\boldsymbol{P}_i)_1^k$.

Note that

$$
\begin{aligned}
(\boldsymbol{v}\boldsymbol{P}_i)_1^k - (\boldsymbol{v}'\boldsymbol{P}_i)_1^k &= \sum_{j=1}^M (v_j - v_j)(\boldsymbol{P}_i(j))_1^k \\
&= \left( \sum_{j'=1}^M (v_{j'} - v'_{j'}) \right)(\boldsymbol{P}_i(M))_1^k + \sum_{j=1}^{M-1}\left( \sum_{j'=1}^j (v_{j'} - v'_{j'}) \right)(\boldsymbol{P}_i(j) - \boldsymbol{P}_i(j+1))_1^k \\
&= 0 + \sum_{j=1}^{M-1}\left( \sum_{j'=1}^j (v_{j'} - v'_{j'}) \right)(\boldsymbol{P}_i(j) - \boldsymbol{P}_i(j+1))_1^k \\
&= 0 + \sum_{j=1}^{M-1}(\boldsymbol{v} - \boldsymbol{v})_1^j(\boldsymbol{P}_i(j) - \boldsymbol{P}_i(j+1))_1^k \\
&\geq 0, \quad\quad\quad\quad\quad\quad\quad\quad\quad\quad\quad\quad\quad\quad\quad\quad\quad\quad (2)
\end{aligned}
$$

where Eq. (2) is because that $(\boldsymbol{v} - \boldsymbol{v})_1^j \geq 0$ and $(\boldsymbol{P}_i(j) - \boldsymbol{P}_i(j+1))_1^k \geq 0$, since we have that $\boldsymbol{v} \gtrsim \boldsymbol{v}'$ and $\boldsymbol{P}_i(j) \gtrsim \boldsymbol{P}_i(j+1)$ by Lemma 4. $\square$

**Lemma 7.** *Under Assumptions 2, 3, 4 and conditioning on event $\mathcal{E}$, for any arm $i$. any integer $\tau \geq 0$ and any probability vectors $\boldsymbol{v}$ and $\boldsymbol{v}'$ such that $\boldsymbol{v} \gtrsim \boldsymbol{v}'$, we have that $\boldsymbol{v}(\boldsymbol{P}'_i)^\tau \gtrsim \boldsymbol{v}'(\boldsymbol{P}_i)^\tau$.*

*Proof.* This is given by applying Lemmas 5 and 6 together. $\square$

Based on these lemmas, we provide the proof of Lemma 1 here.

**Lemma 1.** *Conditioning on event $\mathcal{E}$, we have $\mu(\pi^*, \mathcal{R}) \leq \mu(\pi^{*\prime}, \mathcal{R}')$. Here*

$$
\mathcal{E} = \{\forall i, |j - k| \leq 1, |P_i(j,k) - \hat{P}_i(j,k)| \leq rad(T), |r(i,k) - \hat{r}(i,k)| \leq rad(T)\}.
$$

*Proof.* The key proof idea is to simulate policy $\pi^*$, i.e., emulate it as close as possible in $\mathcal{R}'$, using a fictitious policy $\hat{\pi}^*$ shown in Algorithm 2 , where $\boldsymbol{e}_k$ denotes the probability vector with the $k$-th term equals to 1.

At the beginning, $\hat{\pi}^*$ chooses the same action $i$ as $\pi^*$ does. However, since $\mathcal{R}$ and $\mathcal{R}'$ have different parameters (hence different transitions), the observed state $s'_i(t)$ in $\mathcal{R}'$ does not follow the same distribution as the observed state $s_i(t)$ in $\mathcal{R}$. Thus, to carry out the simulation, we need to record not only the actual observed state $s'_i(t)$, but also a virtual state $s_i(t)$ which follows the same distribution as choosing action $i$ in $\mathcal{R}$. The virtual state $s_i(t)$ and the actual state $s'_i(t)$ are used to update the virtual belief state $z$ and the actual belief state $z'$, respectively. Then, we can pretend to observe the virtual state $s_i(t)$ in our policy $\hat{\pi}^*$ to imitate the trajectory of applying policy $\pi^*$ in problem $\mathcal{R}$ precisely. Specifically, in our simulated policy $\hat{\pi}^*$, we choose the next action according to the virtual

**Algorithm 3** Correspond$(i, \boldsymbol{v}, \boldsymbol{v}', k)$

1: $start = \sum_{j=1}^{k-1} v'_j, end = \sum_{j=1}^{k} v'_j.$
2: **for all** $j$ **do**
3:      $p_j = \sum_{j'=1}^{j} v_{j'}$
4:      **if** $p_j < start$ **then**
5:          $q_j = 0.$
6:      **else**
7:          $q_j = \frac{\min\{p_{j+1}, end\} - p_j}{start - end}.$
8:      **end if**
9: **end for**
10: **Return** $j$ with probability $q_j$.

---

belief state $z = \{(s_i, \tau_i)\}_{i=1}^{N}$ instead of the actual belief state $z' = \{(s'_i, \tau'_i)\}_{i=1}^{N}$. For each time slot $t$, after we record the virtual state $s_i(t)$, we also construct a corresponding virtual reward $r(i, s_i(t))$, while the actual received reward is $r'(i, s'_i(t))$. Since the virtual belief state $z$ follows the trajectory of applying $\pi^*$ in $\mathcal{R}$ precisely, the cumulative virtual reward of applying $\hat{\pi}^*$ in $\mathcal{R}'$ is the same as the cumulative reward of applying $\pi^*$ in $\mathcal{R}$.

We then prove that conditioning on event $\mathcal{E}$, at any time, if we select action $i$, the observed state $s'_i(t)$ and the virtual state $s_i(t)$ satisfies $s_i(t) \geq s'_i(t)$. We use induction to prove it. At the beginning, we have that $s_i = s'_i$ for any $i$.

When we choose to pull arm $i$ at time $t$, suppose that there are $\tau_i$ rounds after the last pull of arm $i$, the last real state of arm $i$ is $s'_i$, and the last virtual state of arm $i$ is $s_i$. If our claim holds at time $t - 1$, i.e., $s_i \geq s'_i$, then by Lemma 7, we know that $\boldsymbol{e}_{s'_i}(\boldsymbol{P}'_i)^{\tau_i} \gtrsim \boldsymbol{e}_{s_i}(\boldsymbol{P}_i)^{\tau_i}$. Denote $\boldsymbol{v}' = \boldsymbol{e}_{s'_i}(\boldsymbol{P}'_i)^{\tau_i}$ and $\boldsymbol{v} = \boldsymbol{e}_{s_i}(\boldsymbol{P}_i)^{\tau_i}$ as the two input vectors of Correspond (line 5 in Algorithm 2), then the Correspond procedure in Algorithm 3, which is for generating an transition that follows distribution $\boldsymbol{v}$, always returns $j \geq k$ since $p_{k-1} \leq start$ in line 4. Thus, the returned virtual state $s_i(t)$ and the actual observed state $s'_i(t)$ satisfies $s_i(t) \geq s'_i(t)$. Thus we finish the proof of the claim.

Since the virtual state $s_i(t)$ follows the distribution under problem instance $\mathcal{R}$, and the next action only depends on the virtual belief state $z$, we know that the cumulative virtual reward is the same as the cumulative reward of applying $\pi^*$ under $\mathcal{R}$.

As for the real reward, we know that conditioning on event $\mathcal{E}$, $r'(i, s'_i(t)) \geq r(i, s'_i(t)) \geq r(i, s_i(t))$. Thus the cumulative real reward of applying $\hat{\pi}^*$ under $\mathcal{R}'$ is larger than the cumulative virtual reward, which equals to the cumulative reward of applying $\pi^*$ under $\mathcal{R}$. This implies that $\mu(\hat{\pi}^*, \mathcal{R}') \geq \mu(\pi^*, \mathcal{R})$.

On the other hand, since $\pi^{*\prime}$ is the best policy of $\mathcal{R}'$, we must have $\mu(\hat{\pi}^*, \mathcal{R}') \leq \mu(\pi^{*\prime}, \mathcal{R}')$. Thus we finish the proof of $\mu(\pi^*, \mathcal{R}) \leq \mu(\pi^{*\prime}, \mathcal{R}')$. $\qquad\square$

## A.2 Proof of Lemma 2

In the following, we only consider the case when all actions $i > 0$ are pulled infinitely often. Since a finitely pulled often action $i > 0$ can be replaced by action $0$ with only a constant regret, we can safely ignore these actions.

**Lemma 8.** *Conditioning on event $\mathcal{E}$, we have that $||\boldsymbol{P}_i(k) - \boldsymbol{P}'_i(k)||_\infty \leq 2rad(T)$ and $|r(i, k) - r'(i, k)| \leq 2rad(T)$.*

*Proof.* Since conditioning on event $\mathcal{E}$ we have that $|\hat{P}_i(j, k) - P_i(j, k)| \leq rad(T)$ and $|\hat{P}_i(j, k) - P'_i(j, k)| \leq rad(T)$, we know that

$$|P'_i(j, k) - P_i(j, k)| \leq |\hat{P}_i(j, k) - P_i(j, k)| + |\hat{P}_i(j, k) - P'_i(j, k)| \leq 2rad(T).$$

Similarly we can prove that $|r(i, k) - r'(i, k)| \leq 2rad(T)$. $\qquad\square$

**Lemma 9.** *Conditioning on event $\mathcal{E}$, for any $\tau, i, k$ and probability vector $\boldsymbol{v}$, we have that $||\boldsymbol{v}(\boldsymbol{P}_i)^\tau - \boldsymbol{v}(\boldsymbol{P}'_i)^\tau||_\infty \leq 2\tau \cdot rad(T)$.*

*Proof.* Note that

$$
\begin{aligned}
||\boldsymbol{v}(\boldsymbol{P}_i)^\tau - \boldsymbol{v}(\boldsymbol{P}_i')^\tau||_\infty &\leq \sum_{\tau'=0}^{\tau-1} ||\boldsymbol{v}(\boldsymbol{P}_i')^{\tau'}(\boldsymbol{P}_i)^{\tau-\tau'-1}(\boldsymbol{P}_i - \boldsymbol{P}_i')||_\infty \\
&= \sum_{\tau'=0}^{\tau-1} ||\boldsymbol{v}(\tau')\boldsymbol{P}_i - \boldsymbol{v}(\tau')\boldsymbol{P}_i'||_\infty \\
&\leq \tau \cdot (2rad(T)),
\end{aligned}
\tag{3}
$$

where $\boldsymbol{v}(\tau') = \boldsymbol{v}(\boldsymbol{P}_i')^{\tau'}(\boldsymbol{P}_i)^{\tau-\tau'-1}$, and Eq. (3) is because that by Lemma 8 we always have $||\boldsymbol{v}(\tau')\boldsymbol{P}_i - \boldsymbol{v}(\tau')\boldsymbol{P}_i'||_\infty \leq 2rad(T)$. $\qquad\square$

**Lemma 10.** *Under Assumptions 2, 4 and conditioning on event $\mathcal{E}$, for any $\tau > \frac{\log T}{\log(1/\lambda_{\max})}, i, k$ and probability vector $v$, we have that $||\boldsymbol{v}(\boldsymbol{P}_i)^\tau - \boldsymbol{v}(\boldsymbol{P}_i')^\tau||_\infty \leq \frac{2M}{1-\lambda_{\max}} \cdot rad(T) + \frac{2M}{T}$.*

*Proof.* Since under Assumptions 2, 4, both $M_i$ (with transition matrix $\boldsymbol{P}_i$) and $M_i'$ (with transition matrix $\boldsymbol{P}_i'$) are ergodic and have unique stationary distributions. After $\frac{\log T}{\log(1/\lambda_{\max})}$ time steps, both the two Markov chains converge to their stationary distributions. By results in [39, 12], conditioning on event $\mathcal{E}$, their stationary distributions satisfy that $\lim_{\tau\to\infty} ||\boldsymbol{v}(\boldsymbol{P}_i)^\tau - \boldsymbol{v}(\boldsymbol{P}_i')^\tau||_\infty \leq \frac{M}{1-\lambda_{\max}} \cdot 2rad(T)$. On the other hand, after $\frac{\log T}{\log(1/\lambda_{\max})}$ steps, the gap between $\boldsymbol{v}(\boldsymbol{P}_i)^\tau$ and the stationary distribution is upper bounded by $\frac{M}{T}$ (similar for $\boldsymbol{v}(\boldsymbol{P}_i')^\tau$) [38].

Thus $||\boldsymbol{v}(\boldsymbol{P}_i)^\tau - \boldsymbol{v}(\boldsymbol{P}_i')^\tau||_\infty \leq \frac{2M}{1-\lambda_{\max}} \cdot rad(T) + \frac{2M}{T}$ for any $\tau > \frac{\log T}{\log(1/\lambda_{\max})}, i, k$, and probability vector $\boldsymbol{v}$. $\qquad\square$

**Lemma 11.** *Under Assumptions 2, 4 and conditioning on event $\mathcal{E}$, for any $\tau, i, k$ and probability vector $v$, we have that $||\boldsymbol{v}(\boldsymbol{P}_i)^\tau - \boldsymbol{v}(\boldsymbol{P}_i')^\tau||_\infty \leq \left( \frac{2\log T}{\log(1/\lambda_{\max})} + \frac{2M}{1-\lambda_{\max}} \right) \cdot rad(T) + \frac{2M}{T}$.*

*Proof.* This lemma is given by directly applying Lemmas 9 and 10. $\qquad\square$

Based on the results in Lemma 11, we denote $gap(T) = \left( \frac{2\log T}{\log(1/\lambda_{\max})} + \frac{2M}{1-\lambda_{\max}} \right) \cdot rad(T) + \frac{2M}{T}$, which will be used frequently in the following analysis.

**Lemma 12.** *Under Assumptions 2, 4 and conditioning on event $\mathcal{E}$, we have that $r(z) - r'(z) \leq 2rad(T) + M \cdot gap(T)$, where $z = \{(\tau_i, s_i)\}_{i=1}^N$ is the belief state of the game, $r(z)$ and $r'(z)$ is the expected given reward of belief state $z$ in problem $\mathcal{R}$ and $\mathcal{R}'$ respectively.*

*Proof.* Note that $r(z) = \sum_{k=1}^m r(i,k)v_k$ and $r'(z) = \sum_{k=1}^m r'(i,k)v_k'$, where $i$ is the chosen action at belief state $z = \{(\tau_i, s_i)\}_{i=1}^N$, and $\boldsymbol{v} = \boldsymbol{e}_{s_i}(\boldsymbol{P}_i)^{\tau_i}$, $\boldsymbol{v}' = \boldsymbol{e}_{s_i}(\boldsymbol{P}_i')^{\tau_i}$.

Thus

$$
\begin{aligned}
|r(z) - r'(z)| &= |\sum_{k=1}^M r(i,k)v_k - \sum_{k=1}^M r'(i,k)v_k'| \\
&= |\sum_{k=1}^M (r(i,k) - r'(i,k))v_k + \sum_{k=1}^M r'(i,k)(v_k - v_k')| \\
&\leq |\sum_{k=1}^M (r(i,k) - r'(i,k))v_k| + |\sum_{k=1}^M r'(i,k)(v_k - v_k')| \\
&\leq 2rad(T) + M \cdot gap(T).
\end{aligned}
\tag{4}
$$

where Eq. (4) is because that $|r(i,k) - r'(i,k)| \leq 2rad(T)$ (Lemma 8) and $|v_k - v_k'| \leq gap(T)$ (Lemma 11). $\qquad\square$

Let $V_t(z)$ denote the expected cumulative reward that we start at belief state $z = \{\tau_i, s_i\}_{i=1}^N$ and implement policy $\pi^{*\prime}$ for $t$ time steps in $\mathcal{R}$, and $\boldsymbol{V}_t$ denote the vector of $V_t(z)$. Similarly, let $V_t'(z)$ be the expected cumulative reward that we start at belief state $z$ and implement $\pi^{*\prime}$ for $t$ times in $\mathcal{R}'$ and $\boldsymbol{V}_t'$ be the vector of $V_t'(z)$. Then we know that $\mu(\pi^{*\prime}, \mathcal{R}') - \mu(\pi^{*\prime}, \mathcal{R}) \leq \lim_{t\to\infty} \frac{1}{t} \|\boldsymbol{V}_t - \boldsymbol{V}_t'\|_\infty$.

Notice that under the fixed policy $\pi^{*\prime}$, $\mathcal{R}$ and $\mathcal{R}'$ are two Markov Chains. Let $\mathcal{T}$ and $\mathcal{T}'$ be the transition matrices of these two Markov Chains, and let $\boldsymbol{r}$ and $\boldsymbol{r}'$ to be the reward vectors of them, respectively. Then, $\boldsymbol{V}_t = \sum_{\tau=0}^{t-1} \mathcal{T}^\tau \boldsymbol{r}$, and $\boldsymbol{V}_t' = \sum_{\tau=0}^{t-1} (\mathcal{T}')^\tau \boldsymbol{r}'$, which implies that

$$
\begin{aligned}
\boldsymbol{V}_{t+1} - \boldsymbol{V}_{t+1}' &= (\mathcal{T}\boldsymbol{V}_t + \boldsymbol{r}) - (\mathcal{T}'\boldsymbol{V}_t' + \boldsymbol{r}') \\
&= \mathcal{T}\boldsymbol{V}_t - \mathcal{T}'\boldsymbol{V}_t' + (\boldsymbol{r} - \boldsymbol{r}') \\
&= \mathcal{T}(\boldsymbol{V}_t - \boldsymbol{V}_t') + (\mathcal{T} - \mathcal{T}')\boldsymbol{V}_t' + (\boldsymbol{r} - \boldsymbol{r}').
\end{aligned}
$$

Since $\mathcal{T}$ represents a transition matrix, $\|\mathcal{T}(\boldsymbol{V}_t - \boldsymbol{V}_t')\|_\infty \leq \|\boldsymbol{V}_t - \boldsymbol{V}_t'\|_\infty$. This implies that

$$
\frac{1}{t}\|\boldsymbol{V}_t - \boldsymbol{V}_t'\|_\infty \leq \frac{1}{t}\sum_{\tau=0}^{t-1} \|(\mathcal{T} - \mathcal{T}')\boldsymbol{V}_\tau'\|_\infty + \frac{1}{t}\sum_{\tau=0}^{t-1} \|\boldsymbol{r} - \boldsymbol{r}'\|_\infty. \tag{5}
$$

Lemma 12 shows that $\frac{1}{t}\sum_{\tau=0}^{t-1} \|\boldsymbol{r} - \boldsymbol{r}'\|_\infty \leq 2rad(T) + M \cdot gap(T)$.

As for the first term in Eq. (5), i.e., $\|(\mathcal{T} - \mathcal{T}')\boldsymbol{V}_\tau'\|_\infty$, notice that for any belief state $z = \{(s_i, \tau_i)\}_{i=1}^N$ that we select action $i \neq 0$, there are only $M$ non-zero values in $\mathcal{T}$ and $\mathcal{T}'$, i.e., transitions to belief states $z_1', \cdots, z_M'$, where $z_k'$ is given by substitute $(s_i, \tau_i)$ by $(k, 1)$, and other actions $i' \neq i$ will add 1 to their $\tau_{i'}$'s. Thus, the $z$-th term in $(\mathcal{T} - \mathcal{T}')\boldsymbol{V}_\tau'$ equals to $\sum_{k=1}^m (v_k(z) - v_k'(z))V_\tau'(z_k')$, where $\boldsymbol{v}(z)$ and $\boldsymbol{v}'(z)$ are probability distributions of the next observed state in $\mathcal{R}$ and $\mathcal{R}'$ under the belief state $z$. Under the event $\mathcal{E}$, $v_k(z) - v_k'(z)$ can be bounded by $\tilde{O}(\frac{M}{1-\lambda_{\max}} rad(T))$ (Lemma 11). On the other hand, since $\sum_{k=1}^M v_k(z) - v_k'(z) = 0$, we have that $\sum_{k=1}^m (v_k(z) - v_k'(z))V_\tau'(z_k') \leq \tilde{O}(\frac{M}{1-\lambda_{\max}} rad(T)) \cdot M \max_{j>k} |V_\tau'(z_j') - V_\tau'(z_k')|$. Therefore, the remaining issue is to bound $\max_{j>k} |V_\tau'(z_j') - V_\tau'(z_k')|$ for any belief state $z$.

Thus, in the following, we concentrate on a fixed tuple $(z, j, k)$, and use the idea of simulated policy again. The simulated policy denoted by $\pi_{z,j,k}^{*\prime}$ is shown in Algorithm 4. Specifically, we pretend to observe a virtual state $s_i = k$ at the beginning, while the actual observed state is $s_i' = j$. Similar as Algorithm 2, in every time step, we observe the real state $s_{a(t)}'(t)$ and real reward $r'(a(t), s_{a(t)}'(t))$, but we also record a virtual state $s_{a(t)}(t)$ and a virtual reward $r'(a(t), s_{a(t)}(t))$, and the next action only depends on the virtual belief state, but not the real belief state. We use $V_\tau^{z,j,k'}(z_j')$ to denote the expected cumulative reward of applying $\pi_{z,j,k}^{*\prime}$ in $\mathcal{R}'$ starting at belief state $z_j'$ (the cumulative real reward). Similarly as before, in $\pi_{z,j,k}^{*\prime}$, $V_\tau'(z_k')$ equals to the cumulative virtual reward.

Then we can bound the gap between $V_\tau^{z,j,k'}(z_j')$ and $V_\tau'(z_k')$ in the next lemma.

**Lemma 13.** *Under Assumptions 2, 4 and conditioning on event $\mathcal{E}$,*

$$
|V_\tau^{z,j,k'}(z_j') - V_\tau'(z_k')| \leq \frac{2M}{1-\lambda_{\max}}.
$$

*Proof.* Recall that $V_\tau^{z,j,k'}(z_j')$ is the expected cumulative reward of applying $\pi_{z,j,k}^{*\prime}$ (Algorithm 4) in $\mathcal{R}'$ and start at belief state $z_j'$. Similar with the previous analysis, in Algorithm 4, the cumulative virtual reward is $V_\tau'(z_k)$.

If we do not pull arm $i$ (the arm chosen in belief state $z$) at time $t$, then the real reward equals to the virtual reward in this time step, since they are under the same problem instance $\mathcal{R}'$. Thus the difference between real reward and virtual reward only appears when we pull arm $i$.

At the beginning, the probability distribution of $s_i$ (the state of arm $i$ in belief state $y$) is $\boldsymbol{e}_k$ and the probability distribution of $s_i'$ (the state of arm $i$ in belief state $y'$) is $\boldsymbol{e}_j$. After $\tau_i$ time steps, the probability distribution of the next virtual state of arm $i$ becomes $\boldsymbol{e}_k(\boldsymbol{P}_i')^{\tau_i}$ and the probability distribution of next real state of arm $i$ becomes $\boldsymbol{e}_j(\boldsymbol{P}_i')^{\tau_i}$. Denote $\boldsymbol{v}(\tau_i) = \boldsymbol{e}_k(\boldsymbol{P}_i')^{\tau_i}$ and $\boldsymbol{v}'(\tau_i) =$

**Algorithm 4** Simulated $\pi_{z,j,k}^{*}{}'$ start at state $z_j'$ based on $\pi^{*\prime}$

---

1: **Init:** The real belief state $y' = z_j'$, and set the virtual belief state $y = z_k'$, let $i$ be the chosen action under $z$.
2: **while true do**
3:      Choose action $a(t)$ as $\pi^{*\prime}$ choose under $y$, and observes state $s'_{a(t)}(t)$, reward $r'(a(t), s'(t))$.
4:      **if** $a(t) \neq i$ **then**
5:          Update the $a(t)$-th term in $y$ and $y'$ to be $(s'_{a(t)}(t), 1)$. For other terms $j \neq a(t)$, set $\tau_j = \tau_j + 1$ and $\tau_j' = \tau_j' + 1$. Observes virtual reward $r'(a(t), s'_{a(t)}(t))$.
6:      **else**
7:          Set $\boldsymbol{v}' = \boldsymbol{e}_{s'_{a(t)}}(\boldsymbol{P}'_{a(t)})^{\tau_{a(t)}}$ and $\boldsymbol{v} = \boldsymbol{e}_{s_{a(t)}}(\boldsymbol{P}'_{a(t)})^{\tau_{a(t)}}$.
8:          Update the $a(t)$-th term in $z$ to be $(s_{a(t)}(t), 1)$, where $s_{a(t)}(t) = $ Correspond$(a(t), \boldsymbol{v}, \boldsymbol{v}', s'_{a(t)}(t))$. For other terms $j \neq a(t)$, set $\tau_j = \tau_j + 1$. Observes virtual reward $r'(a(t), s_{a(t)}(t))$.
9:          Update the $a(t)$-th term in $z'$ to be $(s'_{a(t)}(t), 1)$, for other terms $j \neq a(t)$, set $\tau_j = \tau_j + 1$.
10:      **end if**
11: **end while**

---

$\boldsymbol{e}_j(\boldsymbol{P}'_i)^{\tau_i}$ Then we can upper bound the expected gap between real reward and virtual reward at time $\tau_i$ as $\sum_{\ell=1}^{M} |v_\ell(\tau_i) - v'_\ell(\tau_i)| r'(i, \ell) \leq ||\boldsymbol{v}(\tau_i) - \boldsymbol{v}'(\tau_i)||_1$.

Under Assumptions 2, 4, $\boldsymbol{V} = diag(1, \sqrt{\frac{P'_i(2,1)}{P'_i(1,2)}}, \sqrt{\frac{P'_i(2,1)P'_i(3,2)}{P'_i(1,2)P'_i(2,3)}}, \cdots, \sqrt{\prod_{\ell=1}^{M-1} \frac{P'_i(\ell+1,\ell)}{P'_i(\ell,\ell+1)}})$ satisfies that $\boldsymbol{V}^{-1}\boldsymbol{P}'_i\boldsymbol{V}$ is a symmetric matrix. Thus, the maximum Jordan block size of the Jordan normal form of $\boldsymbol{P}'_i$ equals to 1. According to Fact 3 in [38], such $\boldsymbol{P}'_i$ satisfies that the value of $||\boldsymbol{v}(\tau_i) - \boldsymbol{d}^{(i)'}||_1$ converges to 0 exponentially with rate at most $\lambda^{i'}$, where $\boldsymbol{d}^{(i)'}$ is the unique stationary distribution vector of $M_i$ in $\mathcal{R}'$, and $\lambda^{i'}$ is the absolute value of the second largest eigenvalue of $\boldsymbol{P}'_i$.

Specifically, we have that:

$$
\begin{aligned}
||\boldsymbol{v}(\tau_i) - \boldsymbol{d}^{(i)'}||_1 &\leq M(\lambda^{i'})^{\tau_i} ||\boldsymbol{v}(0) - \boldsymbol{d}^{(i)'}||_1 \\
&\leq 2M(\lambda_{\max})^{\tau_i}.
\end{aligned}
$$

Thus, the $|V_\tau^{z,j,k'}(z_j') - V_\tau'(z_k')|$ is upper bounded by $2M \sum_{\tau_i=0}^{\infty} (\lambda_{\max})^{\tau_i} \leq \frac{2M}{1-\lambda_{\max}}$.      $\square$

**Lemma 14.** *Under Assumptions 2, 4, if all the arms (expected for the default one) are pulled infinitely often, and applying $\pi^{*\prime}$ on $\mathcal{R}'$ is aperiodic, then the stationary distribution of applying $\pi_{z,j,k}^{*}{}'$ on $\mathcal{R}'$ and applying $\pi^{*\prime}$ on $\mathcal{R}'$ is the same.*

*Proof.* Note that when applying policy $\pi^{*\prime}$ (or $\pi_{z,j,k}^{*}{}'$), the arm $i$ (the arm chosen by $\pi^{*\prime}$ under belief state $z$) is pulled infinitely often. Thus, we know that for any $T > 0$, there must be a pull of arm $i$ after $T$.

On the other hand, Under Assumptions 2, 4, we know that the Markov Chain $M_i'$ (with transition matrix $\boldsymbol{P}'_i$) exponentially converges to its unique stationary distribution. Thus the probability of $s_i(t) = s_i'(t)$ converges to 1 as time goes to infinity. Once $s_i = s_i'$, we know that after this time slot we always have $y = y'$, i.e., the virtual belief state equals to the real one. Thus, $\pi_{z,j,k}^{*}{}'$ and $\pi^{*\prime}$ are the same policy after this time step. This means that they will converge to the same stationary distribution.      $\square$

**Lemma 15.** *Under Assumptions 2, 4, if all the arms (expected for the default one) are pulled infinitely often, and applying $\pi^{*\prime}$ on $\mathcal{R}'$ is aperiodic, then*

$$
\lim_{n \to \infty} (V_t^{z,j,k'}(z_j') - V_t'(z_j')) \leq 0. \tag{6}
$$

*Proof.* Let's consider the Bellman equations of applying $\pi^{*\prime}$ on $\mathcal{R}'$. Denote $\mu^{*\prime} = \mu(\pi^{*\prime}, \mathcal{R}')$, and $Q(z)$ the Q-value of belief state $z$.

Then by Bellman equations, we have that:

$$
\begin{aligned}
Q(z_j') &= \mathbb{E}[r_1(z_j')] - \mu^{*\prime} + \mathbb{E}[Q(z_1(z_j'))] \\
&= \mathbb{E}[r_1(z_j')] + \mathbb{E}[r_2(z_j')] - 2\mu^{*\prime} + \mathbb{E}[Q(z_2(z_j'))] \\
&= \cdots \\
&= \sum_{\tau=1}^{t} \mathbb{E}[r_\tau(z_j')] - t\mu^{*\prime} + \mathbb{E}[Q(z_t(z_j'))] \\
&= V_t'(z_j') + \mathbb{E}[Q(z_t(z_j'))] - t\mu^{*\prime},
\end{aligned}
$$

where $r_\tau(z_j')$ and $z_\tau(z_j')$ are the random reward and belief state in the $\tau$-th time step of applying $\pi^{*\prime}$ in $\mathcal{R}'$ that starts at $z_j'$, respectively.

Now let's consider the policy $\pi_{z,j,k}^{*}{}'$, since it is not the best policy, we have that:

$$
\begin{aligned}
Q(z_j') &\geq \mathbb{E}[r_1'(z_j')] - \mu^{*\prime} + \mathbb{E}[Q(z_1'(z_j'))] \\
&\geq \mathbb{E}[r_1'(z_j')] + \mathbb{E}[r_2'(z_j')] - 2\mu^{*\prime} + \mathbb{E}[Q(z_2'(z_j'))] \\
&\geq \cdots \\
&\geq \sum_{\tau=1}^{t} \mathbb{E}[r_\tau'(z_j')] - t\mu^{*\prime} + \mathbb{E}[Q(z_t'(z_j'))] \\
&= V_t^{z,j,k}{}'(z_j') + \mathbb{E}[Q(z_t'(z_j'))] - t\mu^{*\prime}
\end{aligned}
$$

where $r_\tau'(z_j')$ and $z_\tau'(z_j')$ are the random reward and belief state in the $\tau$-th time step of applying $\pi_{z,j,k}^{*}{}'$ in $\mathcal{R}'$ that starts at $z_j'$, respectively. The reason that here is greater than or equal to is because that when applying $\pi_{z,j,k}^{*}{}'$, sometimes we do not choose the best action.

Then we have that

$$
V_t'(z_j') + \mathbb{E}[Q(z_t(z_j))] - t\mu^{*\prime} \geq V_t^{z,j,k}{}'(z_j') + \mathbb{E}[Q(z_t'(z_j))] - t\mu^{*\prime}.
$$

When $t \to \infty$, by Lemma 14, we have that $\mathbb{E}[Q(z_t(z_j))] = \mathbb{E}[Q(z_t'(z_j))]$, which implies that $\lim_{t\to\infty}(V_t^{z,j,k}{}'(z_j') - V_t'(z_j')) \leq 0$. $\qquad\square$

**Lemma 16.** *Under Assumptions 2, 4 and conditioning on event $\mathcal{E}$, if all the actions $i > 0$ are pulled infinitely often, then we have that*

$$
\lim_{\tau\to\infty} \max_{j>k} |V_\tau'(z_j') - V_\tau'(z_k')| \leq \frac{2M}{1 - \lambda_{\max}}. \tag{7}
$$

*Proof.* If applying $\pi^{*\prime}$ on $\mathcal{R}'$ is aperiodic, then we can directly apply Lemmas 13 and 15 to get that for any $j > k$, $\lim_{\tau\to\infty} V_\tau'(z_k') - V_\tau'(z_j') \leq \frac{2M}{1-\lambda_{\max}}$. Similarly, we can prove that $\lim_{\tau\to\infty} V_\tau'(z_j') - V_\tau'(z_k') \leq \frac{2M}{1-\lambda_{\max}}$, which finish the proof in this case.

Then we consider the case that applying $\pi^{*\prime}$ on $\mathcal{R}'$ has a constant period larger than 1. Note that in proof of Lemma 15, we only require that $z_t'(z_j')$ and $z_t(z_j')$ has the same distribution when $t \to \infty$. On the other hand, $z_t'(z_j')$ and $z_t(z_j')$ start from the same state $z_j'$. Thus they are in the same set of states during one period. Because of this, even if applying $\pi^{*\prime}$ on $\mathcal{R}'$ has a constant period larger than 1, $z_t(z_j')$ and $z_t'(z_j')$ converges to the same distribution when $n \to \infty$. This implies that the result in Lemma 15 is still correct. Along with Lemma 13, we know that $\lim_{\tau\to\infty} \max_{j>k} |V_\tau'(z_k') - V_\tau'(z_j')| \leq \frac{2M}{1-\lambda_{\max}}$ still holds. $\qquad\square$

Based on these lemmas, we propose the proof of Lemma 2 here.

**Lemma 2.** *Conditioning on event $\mathcal{E}$, $\mu(\pi^{*\prime}, \mathcal{R}') - \mu(\pi^{*\prime}, \mathcal{R}) = \tilde{O}\left(\frac{M^3}{(1-\lambda_{\max})^2} T^{-\frac{1}{3}}\right)$.*

*Proof.* Eq. (5) shows that $\mu(\pi^{*\prime}, \mathcal{R}') - \mu(\pi^{*\prime}, \mathcal{R}) \le \lim_{t\to\infty} \frac{1}{t}\sum_{\tau=0}^{t-1} ||(\mathcal{T} - \mathcal{T}')V'_\tau||_\infty + \lim_{t\to\infty} \frac{1}{t}\sum_{\tau=0}^{t-1} ||\boldsymbol{r} - \boldsymbol{r}'||_\infty$.

Lemma 12 shows that $\lim_{t\to\infty} \frac{1}{t}\sum_{\tau=0}^{t-1} ||\boldsymbol{r} - \boldsymbol{r}'||_\infty \le 2rad(T) + M \cdot gap(T)$.

As for $||(\mathcal{T} - \mathcal{T}')V'_\tau||_\infty$, note that for any belief state $z = \{(s_i, \tau_i)\}_{i=1}^N$ that we select action $i \ne 0$, there are only $M$ non-zero values in $\mathcal{T}$ and $\mathcal{T}'$, i.e., transitions to belief states $z'_1, \cdots, z'_M$, where $z'_k$ is given by substituting $(s_i, \tau_i)$ by $(k, 1)$, and other actions $j \ne i$ will increase their $\tau_j$ values by 1. Thus, the $z$-th term in $(\mathcal{T} - \mathcal{T}')V'_\tau$ equals to $\sum_{k=1}^m (v_k(z) - v'_k(z))V'_\tau(z'_k)$, where $\boldsymbol{v}(z)$ and $\boldsymbol{v}'(z)$ are probability distributions of the next observed state in $\mathcal{R}$ and $\mathcal{R}'$ under the belief state $z$. That is,

$$
\begin{aligned}
\lim_{t\to\infty} ||(\mathcal{T} - \mathcal{T}')V'_\tau||_\infty &= \lim_{t\to\infty} \sum_{k=1}^m (v_k(z) - v'_k(z))V'_\tau(z'_k) \\
&\le \lim_{t\to\infty} \frac{M}{2} ||\boldsymbol{v}(z) - \boldsymbol{v}'(z)||_\infty \max_{j>k} |V'_\tau(z'_j) - V'_\tau(z'_k)| \qquad (8) \\
&\le \frac{M}{2} gap(T) \lim_{t\to\infty} \max_{j>k} |V'_\tau(z'_j) - V'_\tau(z'_k)| \\
&\le \frac{M^2}{1 - \lambda_{\max}} gap(T), \qquad (9)
\end{aligned}
$$

where Eq. (8) is because that $\boldsymbol{v}(z)$ and $\boldsymbol{v}'(z)$ are probability vectors with dimension $M$, and Eq. 9 comes from Lemma 16.

Thus, $\lim_{t\to\infty} \frac{1}{t}\sum_{\tau=0}^{t-1} ||(\mathcal{T} - \mathcal{T}')V'_\tau||_\infty \le \frac{M^2}{1-\lambda_{\max}} gap(T)$. Since $gap(T) = \tilde{O}\left(\frac{M}{1-\lambda_{\max}} rad(T)\right)$, we have that $\mu(\pi^{*\prime}, \mathcal{R}') - \mu(\pi^{*\prime}, \mathcal{R}) \le \tilde{O}\left(\frac{M^3}{(1-\lambda_{\max})^2} rad(T)\right) + 2rad(T) + \tilde{O}\left(\frac{M^2}{1-\lambda_{\max}} rad(T)\right) = \tilde{O}\left(\frac{M^3}{(1-\lambda_{\max})^2} rad(T)\right)$.

Note that when $m(T) = T^{\frac{2}{3}}$, we have that $rad(T) = \tilde{O}(T^{-\frac{1}{3}})$, therefore $\mu(\pi^{*\prime}, \mathcal{R}') - \mu(\pi^{*\prime}, \mathcal{R})$ is upper bounded by $\tilde{O}\left(\frac{M^3}{(1-\lambda_{\max})^2} T^{-\frac{1}{3}}\right)$. $\qquad\square$

### A.3 Other Lemmas and Facts

**Fact 1.** *The length $T_1$ of the exploration phase satisfies that*

$$
\mathbb{E}[T_1] = \tilde{O}\left(\frac{N}{d_{\min}} m(T)\right).
$$

**Lemma 17.** *With probability at least $1 - \frac{8NM}{T}$, $\mathcal{E}$ holds.*

*Proof.* Recall that

$$
\mathcal{E} = \{\forall i, |j - k| \le 1, |P_i(j, k) - \hat{P}_i(j, k)| \le rad(T), |r(i, k) - \hat{r}(i, k)| \le rad(T)\}.
$$

For any action $i$ and state $j, k$, we have that

$$
\begin{aligned}
\Pr[|P_i(j, k) - \hat{P}_i(j, k)| \ge rad(T)] &\le 2\exp(-2m(T)(rad(T))^2) \qquad (10) \\
&\le 2\exp\left(-2m(T) \cdot \frac{\log T}{2m(T)}\right) \\
&\le \frac{2}{T},
\end{aligned}
$$

where Eq. (10) is given by Chernoff-Hoeffding inequality [19]. Similarly, $\Pr[|r(i, k) - \hat{r}(i, k)| \le rad(T)] \le \frac{2}{T}$.

Thus, by union bound, $\mathcal{E}$ holds with probability at least $1 - \frac{8NM}{T}$. $\qquad\square$

### A.4 Main Proof of Theorem 1

#### A.4.1 Regret in Exploration Phase

By Fact 1, we know the regret in exploration phase has upper bound $\tilde{O}(\frac{N}{d_{\min}}m(T))$.

#### A.4.2 Regret in Exploitation Phase

Let $T_2$ be the number of time steps in the exploitation phase. Note that applying $\pi^{*\prime}$ in $\mathcal{R}$ has an average reward $\mu(\pi^{*\prime}, \mathcal{R})$. According to results in [6], the cumulative reward of applying policy $\pi^{*\prime}$ in $\mathcal{R}$ for $T_2$ time steps is lower bounded by $T_2\mu(\pi^{*\prime}, \mathcal{R}) - \mathcal{C}$, where $\mathcal{C}$ is the diameter of applying policy $\pi^{*\prime}$ in $\mathcal{R}$, which does not depend on $T$.

Then we can write the upper bound of cumulative regret in exploitation phase as

$$T_2(\mu(\pi^*, \mathcal{R}) - \mu(\pi^{*\prime}, \mathcal{R})) + \mathcal{C}. \tag{11}$$

Since $\mathcal{C}$ is a constant that does not depend on $T$, we can concentrate on the term depends on $T$ (or $T_2$), i.e., $T_2(\mu(\pi^*, \mathcal{R}) - \mu(\pi^{*\prime}, \mathcal{R}))$. To bound this term, we write $\mu(\pi^*, \mathcal{R}) - \mu(\pi^{*\prime}, \mathcal{R})$ as:

$$\mu[\pi^*, \mathcal{R}] - \mu(\pi^{*\prime}, \mathcal{R}) \quad = \quad [\mu(\pi^*, \mathcal{R}) - \mu(\pi^{*\prime}, \mathcal{R}')] + [\mu(\pi^{*\prime}, \mathcal{R}') - \mu(\pi^{*\prime}, \mathcal{R})]. \tag{12}$$

By Lemma 1, conditioning on event $\mathcal{E}$, the first term is upper bounded by $0$.

By Lemma 2, conditioning on event $\mathcal{E}$, the second term is upper bounded by $\tilde{O}\left(\frac{M^3}{(1-\lambda_{\max})^2}T^{-\frac{1}{3}}\right)$.

Therefore, conditioning on event $\mathcal{E}$, we have that

$$T_2(\mu(\pi^*, \mathcal{R}) - \mu(\pi^{*\prime}, \mathcal{R})) \leq \tilde{O}\left(\frac{M^3}{(1-\lambda_{\max})^2}T^{\frac{2}{3}}\right)$$

#### A.4.3 The Total Regret

From the above analysis, the total regret is upper bounded by:

$$
\begin{aligned}
Reg(T) &\leq \mathbb{E}[T_1] + \tilde{O}\left(\frac{M^3}{(1-\lambda_{\max})^2}T^{\frac{2}{3}}\right) + 8NM + \mathcal{C} \\
&\leq \tilde{O}\left(\frac{N}{d_{\min}}m(T)\right) + \tilde{O}\left(\frac{M^3}{(1-\lambda_{\max})^2}T^{\frac{2}{3}}\right) + 8NM + \mathcal{C} \\
&= \tilde{O}\left(\left(\frac{N}{d_{\min}} + \frac{M^3}{(1-\lambda_{\max})^2}\right)T^{\frac{2}{3}}\right).
\end{aligned}
$$

## B Reduced Regret Bound for Colored-UCRL2 or Thompson Sampling

Denote $\boldsymbol{h}'$ the bias vector of applying policy $\pi^{*\prime}$ in $\mathcal{R}'$, and $U = \max_{z,j>k}|h'(z_j') - h'(z_k')|$. The colored-UCRL2 policy [33] and Thompson Sampling policy [22] both achieve regret upper bounds of $O(U\sqrt{T})$, according to their analysis. To bound the value of $U$, they both directly apply an upper bound $D$, which is the diameter of applying policy $\pi^{*\prime}$ in $\mathcal{R}'$, according to [6].

However, note that $|h'(z_j') - h'(z_k')| = \lim_{t\to\infty}|V_t'(z_j') - V_t'(z_k')|$, and Lemma 16 states that $\lim_{t\to\infty}\max_{j>k}|V_t'(z_j') - V_t'(z_k')| \leq \frac{2M}{1-\lambda_{\max}}$ for any $z$. Therefore,

$$
\begin{aligned}
U &= \max_{z,j>k}|h'(z_j') - h'(z_k')| \\
&= \max_{z,j>k}\lim_{t\to\infty}|V_t'(z_j') - V_t'(z_k')| \\
&= \lim_{t\to\infty}\max_{z,j>k}|V_t'(z_j') - V_t'(z_k')| \\
&\leq \frac{2M}{1-\lambda_{\max}}.
\end{aligned}
$$

| (a) Regret: two channels | (b) Regret: three channels |

Figure 2: More experiments: Comparison of regrets of different algorithms

This implies that the regret upper bounds in [33, 22] can be reduced to $O(\frac{M}{1-\lambda_{\max}}\sqrt{T})$, whose constant factor is a polynomial one.

More importantly, in the proof of Lemma 16, we only use Assumptions 2, 4 to make sure that the both the original Markov chain $M_i$ (with transition matrix $\boldsymbol{P}_i$) and the constructed Markov chain $M_i'$ (with transition matrix $\boldsymbol{P}_i'$) are ergodic. Therefore, Lemma 16 is not limited to the setting in this paper, but instead can be applied to more general settings and reduce the exponential factors in the regret upper bounds under other learning policies.

## C   Approximation Oracle

**Theorem 2.** *If* `Oracle` *returns an approximate policy with ratio $\lambda$, then under Assumptions 1, 2, 3 and 4, the approximation regret (with approximate ratio $\lambda$) of Restless-UCB with $m(T) = T^{\frac{2}{3}}$ in an online restless bandit problem $\mathcal{R}$ is upper bounded by $\tilde{O}(T^{\frac{2}{3}})$.*

*Proof.* First, we see that the exploration phase still results in $\tilde{O}(T^{\frac{2}{3}})$ regret. Next, we come to the exploitation phase.

Similarly, the regret in the exploitation phase can be upper bounded by $T_2(\lambda\mu(\pi^*, \mathcal{R}) - \mu(\tilde{\pi}', \mathcal{R}))$. We can similarly write $\lambda\mu(\pi^*, \mathcal{R}) - \mu(\tilde{\pi}', \mathcal{R})$ as

$$\lambda\mu(\pi^*, \mathcal{R}) - \mu(\tilde{\pi}', \mathcal{R}) = [\lambda\mu(\pi^*, \mathcal{R}) - \mu(\tilde{\pi}', \mathcal{R}')] + [\mu(\tilde{\pi}', \mathcal{R}') - \mu(\tilde{\pi}', \mathcal{R})]$$

Note that Lemma 1 still works in this case. Thus, we must have $\mu(\tilde{\pi}', \mathcal{R}') \geq \lambda\mu(\pi^{*\prime}, \mathcal{R}') \geq \lambda\mu(\pi^*, \mathcal{R})$ with high probability. However, Lemma 16 does not work here since it relies on the optimality of $\pi^{*\prime}$.

According to the results in [6], we can use $D$, the diameter of applying $\tilde{\pi}'$ in the problem $\mathcal{R}'$ as an upper bound for $\lim_{\tau\to\infty}\max_{j>k}|V_\tau'(z_k') - V_\tau'(z_j')|$. Thus, the regret in exploitation phase is still $\tilde{O}(T^{\frac{2}{3}})$ while the constant factor here is much larger and probably exponential.

Together with the regret in exploration phase, we finish the proof. $\qquad\square$

Note that although the constant factor in the regret bound can be exponential, the algorithm complexity is still polynomial and much better than the colored-UCRL2 policy [33].

## D   More Experiments on Real Datasets

We also use the dataset in [40] for packet transmission via a wireless link with different frequency channels. Table 3 of [40] provides the transition matrices of the two-state Markov Chains for different

channels. In this setting, a bad state results in a packet loss while a good state guarantees a successful transition. In Figure 2(a), one can use frequency bands of 551MHz and 665MHz. In Figure 2(b), one can use frequency bands of 551MHz, 629MHz and 665MHz.

One can see that Restless-UCB still outperforms other policies. The TS policy suffers from a linear regret, since its support does not contain the real transition matrices, and colored-UCRL2 performs worse than Restless-UCB as well. These results also demonstrate the effectiveness of Restless-UCB.