[Reviews · NeurIPS 2020]

Review 1

Summary and Contributions: This work studies the restless bandit problem. Their main contribution is proposing a explore-then-commit type algorithm. By assuming the underline Markov chain are birth-death processes, they can show that the resulting regret upper bound is polynomial in the size of the problem. Besides, they can also reduce the computational complexity from exponential to polynomial as well.

Strengths: The results are interesting as they can show significant constant reduction in regret. The empirical evaluation also shows the good performance of the proposed algorithm.

Weaknesses: One key assumption is that the underline Markov chain is a birth-death process. Since the chain becomes linear, then it is not suprising that the regret and complexity are reduced to polynomial instead of exponential.

Correctness: The claims and methos are correct.

Clarity: Yes. The paper is well written and organized.

Relation to Prior Work: Yes.

Reproducibility: Yes

Additional Feedback: 1) As I mentioned above, one question is whether the results here can be generalized to other cases in restless bandits. 1.a) Could this model be generalized to infinite state birth-death process? For example, M/M/1 queue in queueing system? 1.b) Could this model be generalized to other stochastic process that are not birth-death process? For example, Moran process? 2) Since the algorithm is an explore-then-commit type algorithm, it is also crucial in the experimental results to show the error bar of regret. Because this type of algorithm usually suffer from large variance in empirical performance and is not analyzed in theory. 3) The regret in Theorem 1 depends on M^3. Is this factor tight enough? Could it be reduced to M^2 or even M? ======= After Rebuttal ====== I think the authors are aware of the concerns about birth-death assumption and the lacking of lower bound on the regret complexity. These are not fully addressed in the rebuttal. So I maintained my score.


Review 2

Summary and Contributions: The paper considers an online restless bandit problem, where each arm’s state evolves over time regardless of the player's actions, according to a birth-death Markov chain, the reward of pulling each arm depends on both the arm and the current state of the pulled arm. A player at each time can only observe the reward and the state of the pulled arm. An online algorithm (Restless-UCB) for learning the optimal policy that maximizes the total expected reward is proposed, which the paper claims to achieve a better regret upper bound (sub-linear) and computation complexity than existing algorithms [28]. The proposed algorithm is of the explore-then-commit type, where the arms are explored for a pre-specified amount of time (o(T)), and then use parameter estimates from the exploration to construct an offline instance of the problem; an oracle is assumed available to general an optimal policy from the offline instance, which is then used indefinitely thereafter. For the special case considered in the paper, birth-death chain with positively correlated state transitions and monotonically decreasing rewards as "birth" progresses, this algorithm achieves sub-linear strong regret (on par but slightly worse) with much lower complexity compared to prior work such as [28].

Strengths: The paper is clearly written and appears technically sound. As a special case of more general restless bandits, it obtains computationally efficient results.

Weaknesses: 1. Comparison with prior work is woefully inadequate. For instance, earlier in the paper (page 2) citing [35, 10, 21] it is stated these prior works have a log(T) regret; later on page 6, citing the same papers, it is stated that these suffer from \Theta(T) regret. A quick glance at these paper reveals a number of differences: (i) these earlier works are for weak regret, comparing with single best action policies, while this submission is for strong regret, comparing with the best dynamic policy, (ii) earlier works [35] are for much more general Markov chains, while this paper has much more limiting assumptions (birth-death chain, positively correlated state transitions, monotonic state rewards), (iii) the earlier works do indeed have log(T) regret, and are generally finite time bound (bound holds uniform in time). Also, it is incorrect to say that these restrictive assumptions are common in the bandit literature (page 4): they are perhaps common in the sub-literature of using bandit models to solve dynamic spectrum access problems, but is not at all common in the general bandit literature, and does not appear to have much support/motivation outside of this area. 2. The more relevant comparison, with [28] is equally problematic and misleading. Again, the assumption of a birth-death chain with positively correlated state transitions and monotonically decreasing rewards makes the present problem a very special case of a general Markov chain, so it is not surprising that one can construct algorithm to take advantage of these special features to reduce its complexity. These are indeed key ideas to this paper (Lemmas 1 & 2) because of these restrictive assumptions; these are not necessarily key ideas that prior works like [28] missed (as the wording would seem to suggest) because the latter was considering a much more general model. It would be highly desirable and more informative for the paper to state clearly that their model is a special case (starting from the paper title), but the assumptions allowed them to obtain better results. 3. In the same context, it would also be very helpful if the paper could explain what happens if one takes the result from [28] and applies it to this special case, rather than directly quoting the results form [28] which obviously is for the more general case.

Correctness: Technically correct, but some misleading statements (see above).

Clarity: Yes.

Relation to Prior Work: Please see earlier comments.

Reproducibility: No

Additional Feedback: - What is the oracle used in the experiments? - Suggest revising the title to make it clear the problem model is not for any restless bandits. ====== post rebuttal ============ The authors have addressed my concern on literature review. The limitation of birth-death chain remains. I have revised my score to 6.


Review 3

Summary and Contributions: This paper studies the online restless bandit problem, where the state of each arm evolves according to a Markov chain, and the reward of pulling an arm depends on both the pulled arm and the current state of the corresponding Markov chain. This work propose is a learning policy that follows the explore-then-commit principle, called restless-UCB. The main challenges of online restless bandit learning include high computation complexity and exponential factor in the regret bound. Corresponding to the challenges, the main contributions of this paper include (1) low regret bound: the proposed algorithm has a regret bound which is only polynomial in the number of arms and states. (2) low computation complexity. Restless bandit solutions usually have high (exponential with respect to N) computation complexity. The proposed algorithm has a linear computation complexity.

Strengths: Strength: (1) The improvement in the computation complexity of restless bandit has great practical meaning in order for the restless bandit to be useful (2) The theoretical analysis developed in this paper is solid and novel.

Weaknesses: Weakness: (1) Birth-death MDP: This paper focuses on the birth-death MDP setting. Although the authors emphasized that Birth-death MDP has many important applications, it would be helpful to discuss a potential extension to the more general (PO)MDP setting. (2) Availability of the oracle: While achieving an O(N) computation complexity, the proposed solutions depends on the availability of a good oracle. The authors should justify a little more about whether such an oracle can be easily available in different settings of the problem (this is also related to weakness 1). (3) State sparsity: one main property used to reduce regret is the sparsity of state transition. However, this assumption is not well justified in the paper.

Correctness: Yes

Clarity: Yes

Relation to Prior Work: Yes but needs further improvments. (1) It will be helpful to compare different assumptions used in existing restless bandit solutions. For example is the state sparsity assumption already exploited in existing literature. (2) Restless bandit is also related with non-stationary bandit. Although not directly comparable, it would be helpful to also discuss the relationship with non-statioanry bandits. In addition, many only statistical tests developed in non-stationary bandit learning literature can potentially be used to estimate or predict the current state of the restless bandit.

Reproducibility: Yes

Additional Feedback: *************************Post rebuttal*********************** My concerns about prior work and the oracle are largely addressed by the authors' responses. I am sticking with my original score (7, accept).

[Author Response · NeurIPS 2020]

**All Reviewers**: Thank you for your effort and the insightful comments! We will revise our paper accordingly.

**Common comment on the setting**: (i) First, we would like to emphasize again that the birth-death Markov setting,
though more restricted, has many applications in areas such as networks, communications and recommendations.
(ii) Second, even under the birth-death setting, our algorithm is the only one with sublinear regret upper bound and
polynomial time complexity. The time complexity of the Colored-UCRL2 in [Ortner et. al., 2012] ([28]) remains
exponential with $N$ under our setting. This is so because Colored-UCRL2 cannot explore the structure of the
Markov chain, and only regards the restless bandit problem as an MDP problem, whose states are the belief states
$z = \{(s_i, \tau_i)\}_{i=1}^N$ (as mentioned in lines 191-198 of our paper). Since the number of states is exponential in $N$, it
means that Colored-UCRL2 need an exponential time complexity to find out the best policy for the MDP, even under
our birth-death setting. This is also demonstrated by our experiment. In Table 1 (page 8 in our paper), we consider
birth-death Markov chains, and the time cost of Colored-UCRL2 grows exponentially as $N$ increases. (iii) Thirdly, our
algorithm and analysis are not limited in the birth-death setting. In particular, Lemma 1 (which ensures low complexity)
holds so long as we have the following property: $\forall i, k, P_i(k) \gtrsim P_i(k+1)$, where $P_i(k)$ represents the transition vector
of arm $i$ under state $k$, and $v \gtrsim v'$ is defined as $\forall k, \sum_{j=1}^k v_j \geq \sum_{j=1}^k v'_j$. Lemma 2 (which reduces the constant factor)
also only requires that the Markov chains are ergodic. This means that it can be applied to reduce the constant factors in
regret upper bounds in general restless bandit problems, such as [Ortner et. al., 2012] ([28]) and [Jung et. al., 2019]
([17]).

In particular, compared with [Ortner et. al., 2012] ([28]), there are two points we want to highlight: (i) even under our
setting, our algorithm is the only one that achieves sublinear regret and polynomial complexity; (ii) our analysis helps
to reduce the constant factor in the regret bounds of [Ortner et. al., 2012] ([28]). We will improve our explanation of the
comparison to make the claims clear and avoid misunderstanding.

**Reviewer 1**

*Q1: Adapting our results on stochastic processes that have infinite states or do hot have a birth-death structure.* We
can adapt our algorithm and analysis as long as the Markov chains $M_i$'s are ergodic and have the property that $\forall i, k$,
$P_i(k) \gtrsim P_i(k+1)$. For example, consider a discrete queueing system. In each time slot, there is at most one arrival
(with probability $\lambda$) and similarly at most one departure (with probability $\mu$), so that the corresponding Markov chain $M$
has a birth-death structure. We also assume that the buffer size is infinity, so that $M$ has infinite number of states. For this
system, we can adapt our algorithm and analysis as long as $1 > \mu > \lambda > 0$. This is because that i) $M$ is positive recurrent
and aperiodic when $1 > \mu > \lambda > 0$; ii) in Markov chain $M$, $P(k, k+1) + P(k+1, k) = \lambda(1-\mu) + (1-\lambda)\mu < 1$,
which implies that $\forall k, P(k) \gtrsim P(k+1)$.

*Q2: About the error bar.* Below are some experiments (Figures 1.(a), 1.(b), 1.(c) and 1.(d) in our paper) with error bar.
We can see that Restless-UCB does not lead to a large variance. On the other hand, the TS policy suffers from a large
variance when $T$ is small. This is due to the high degree of randomness on the samples it draws when $T$ is small.

(a) Figure 1.(a)    (b) Figure 1.(b)    (c) Figure 1.(c)    (d) Figure 1.(d)

*Q3: Is the $M^3$ factor tight enough?* We conjecture that a lower bound will be $\Omega(M^2)$. Formally establishing this result
is an interesting future research topic.

**Reviewers 2 and 4**

*Q4 - Relation to prior work.* Following your helpful suggestions on the weak regret, non-stationary bandits and
assumptions in existing restless bandit solutions, we will including more related works and discuss the relation with
them in details.

*Q5 - About the oracle.* We use the offline policy proposed in [Liu and Zhao, 2010] as the oracle in our experiments,
and we will clarify this in the paper. In particular, [Liu and Zhao, 2010] proposes an offline policy for the case where
all Markov chains only have two states. Note that in addition to the exact oracle, Restless-UCB policy can also be
combined with approximate oracles to maintain low time-complexity, such as [Guha et. al., 2010] and [Liu and Zhao,
2009], while achieving performance guarantees. This is a unique feature not possessed by other existing algorithms.

[Meta-Review · NeurIPS 2020]

I must first admit that judging this paper was a fairly challenging task given the mixed opinions expressed by the reviewers, together with my own impressions after having scrutinized the manuscript in detail. The reviewers largely agree that the paper deserves credit as it tackles the challenging, relevant and (relatively) scarcely studied topic of restless bandit learning. I believe the main value of the paper is in the introduction of the birth-death Markov chain structure for arms of a restless bandit, together with the monotonicity and positive correlation assumptions on rewards and transitions. These are not unnatural assumptions, as evidenced by modeling literature on scheduling over wireless channels and queueing systems, and seem to greatly alleviate the computational complexity of a portion of the learning process. On the other hand, the reviewers are not fully convinced about the significance of the proposed algorithm and regret bound proven in the paper, given that the analysis is carried out for a highly structured ensemble of Markov decision processes. A technical question about the analysis, in this regard, that I was unable to satisfactorily resolve is: Why is optimistic planning actually required in an explore-then-commit type algorithm? At least in bandits, this does not appear to give order-wise improvements in regret. On the subject of the new structural assumptions, while the identification of this structure is, of course, not always an easy task, I feel that a paper introducing new assumptions must do a more comprehensive job at proposing a solution. For instance, the question of how computationally demanding a planning oracle is to implement remains largely unresolved (the author response gives only a superficial insight into it); thus, it is not clear if the exponential reduction in complexity in handling the uncertainty set is finally an advantage in hand (what if the planner is still very expensive to run?). Finally, for a work that proposes to lighten the computational burden of learning restless bandits, it is disappointing to see practical experiments carried out on Markov chains with only 2 states (trivially birth-death Markov chains), whereas a more convincing demonstration would have involved showing an end-to-end algorithmic implementation of a learning algorithm on arms with several states, which would illustrate the full power of this framework. A more comprehensive set of experiments on instances involving many states per arm is highly recommended to demonstrate the value of the algorithmic solution. In summary, the paper is balanced on a knife edge, but I feel that the author(s) could do a more comprehensive job at more solidly motivating the proposed Markov chain structure, explicitly discussing the effort involved in offline planning, shedding more light on optimality within this structured class (perhaps through exposing fundamental limits on regret), and demonstrating experimental results on more complex setups.